# MARS: META-LEARNING AS SCORE MATCHING IN THE FUNCTION SPACE

**Krunoslav Lehman Pavasovic** *
ETH Zurich
Switzerland
klehman@ethz.ch

**Jonas Rothfuss** *
ETH Zurich
Switzerland
rojonas@ethz.ch

**Andreas Krause**
ETH Zurich
Switzerland
krausea@ethz.ch

## ABSTRACT

Meta-learning aims to extract useful inductive biases from a set of related datasets. In Bayesian meta-learning, this is typically achieved by constructing a prior distribution over neural network parameters. However, specifying families of computationally viable prior distributions over the high-dimensional neural network parameters is difficult. As a result, existing approaches resort to meta-learning restrictive diagonal Gaussian priors, severely limiting their expressiveness and performance. To circumvent these issues, we approach meta-learning through the lens of functional Bayesian neural network inference, which views the prior as a stochastic process and performs inference in the function space. Specifically, we view the meta-training tasks as samples from the data-generating process and formalize meta-learning as empirically estimating the law of this stochastic process. Our approach can seamlessly acquire and represent complex prior knowledge by meta-learning the score function of the data-generating process marginals instead of parameter space priors. In a comprehensive benchmark, we demonstrate that our method achieves state-of-the-art performance in terms of predictive accuracy and substantial improvements in the quality of uncertainty estimates.

## 1 INTRODUCTION

Using data from related tasks is of key importance for sample efficiency. Meta-learning attempts to extract prior knowledge (i.e., inductive bias) about the unknown data generation process from these related tasks and embed it into the learner so that it generalizes better to new learning tasks (Thrun & Pratt, 1998; Vanschoren, 2018). Many meta-learning approaches try to amortize or re-learn the entire inference process (e.g., Santoro et al., 2016; Mishra et al., 2018; Garnelo et al., 2018) or significant parts of it (e.g., Finn et al., 2017; Yoon et al., 2018). As a result, they require large amounts of meta-training data and are prone to meta-overfitting (Qin et al., 2018; Rothfuss et al., 2021a).

The Bayesian framework provides a sound and statistically optimal method for inference by combining prior knowledge about the data-generating process with new empirical evidence in the form of a dataset. In this work, we adopt the Bayesian framework for inference at the task level and only focus on meta-learning informative Bayesian priors. Previous approaches (Amit & Meir, 2018; Rothfuss et al., 2021a) meta-learn Bayesian Neural Network (BNN) prior distributions from a set of related datasets; by meta-learning the prior distribution and applying regularization at the meta-level, they facilitate positive transfer from only a handful of meta-training tasks. However, BNNs lack a parametric family of (meta-)learnable priors over the high-dimensional space of neural network (NN) parameters that is both computationally viable and, simultaneously, flexible enough to account for the over-parametrization of NNs. In practice, both approaches use a Gaussian family of priors with a diagonal covariance matrix, which is too restrictive to accurately match the complex probabilistic structure of the data-generating process.

To address these shortcomings, we take a new approach to formulating the meta-learning problem and represent prior knowledge in a novel way. We build on recent advances in functional approximate inference for BNNs that perform Bayesian inference in the function space rather than in the parameter space of neural networks (Wang et al., 2018; Sun et al., 2019). When viewing the BNN prior and posterior as stochastic processes, the perfect Bayesian prior is the (true) data-generating

---

*Equal contribution.

stochastic process itself. Hence, we view the meta-training datasets as samples from the meta-data-generating process and interpret meta-learning as empirically estimating the law of this stochastic process. More specifically, we meta-learn the *score function* of its marginal distributions, which can then directly be used as a source of prior knowledge when performing approximate functional BNN inference on a new target task. This ultimately allows us to use flexible neural network models for learning the score and overcome the issues of meta-learning BNN priors in the parameter space.

In our experiments, we demonstrate that our proposed approach, called *Meta-learning via Attention-based Regularised Score estimation (MARS)*, consistently outperforms previous meta-learners in predictive accuracy and yields significant improvements in the quality of uncertainty estimates. Notably, MARS enables positive transfer from only a handful of tasks while maintaining reliable uncertainty estimates. This promises fruitful future applications to domains like molecular biology or medicine, where meta-training data is scarce and reasoning about epistemic uncertainty is crucial.

## 2  RELATED WORK

**Meta-Learning.**  Common approaches in meta-learning amortize the entire inference process (Santoro et al., 2016; Mishra et al., 2018; Ravi & Beatson, 2018; Garnelo et al., 2018), learn a good neural network initialization (Finn et al., 2017; Rothfuss et al., 2019; Nichol et al., 2018; Kim et al., 2018) or a shared embedding space (Baxter, 2000; Vinyals et al., 2016; Snell et al., 2017). Although these approaches can meta-learn complex inference patterns, they require a large amount of meta-training data and often perform poorly in settings where data is scarce. Another line of work uses a hierarchical Bayesian approach to meta-learn priors over the NN parameters (Pentina & Lampert, 2014; Amit & Meir, 2018; Rothfuss et al., 2021a). Such methods perform much better on small data. However, they suffer from the lack of expressive families of priors for the high-dimensional and complex parameter space of NNs, making too restrictive assumptions to represent complex inductive biases. Our approach overcomes these issues by viewing the problem in the function space and directly learning the score, which can easily be represented by a NN instead of a prior distribution. Also related to our stochastic process approach are methods that meta-learn Gaussian Process (GP) priors (Fortuin et al., 2019; Rothfuss et al., 2021b; 2022). However, the GP assumption is quite limiting, while MARS can, in principle, match the marginals of any data-generating process.

**Score estimation.**  We use score estimation as a central element of our meta-learning method. In particular, we use a parametric approach to score matching and employ an extended version of the score matching objective of Hyvärinen & Dayan (2005). For high-dimensional problems, Song et al. (2020); Pang et al. (2020) propose randomly sliced variations of the score matching loss. Alternatively, there is a body of work on nonparametric score estimation (Canu & Smola, 2006; Liu et al., 2016; Shi et al., 2018; Engl et al., 1996; Zhou et al., 2020). Among those, the Spectral Stein Gradient Estimator (Shi et al., 2018) has been used for estimating the stochastic process marginals for functional BNN inference in a setting where the stochastic prior is user-defined and allows for generating arbitrarily many samples (Sun et al., 2019). Such estimators make it much harder to add explicit dependence on the measurement sets and prevent meta-overfitting via regularization, making them less suited to our problem setting.

## 3  BACKGROUND

**Bayesian Neural Networks.**  Consider a regression task with data $\mathcal{D} = \left(\mathbf{X}^{\mathcal{D}}, \mathbf{y}^{\mathcal{D}}\right)$ that consists of $m$ i.i.d. noisy function evaluations $y_j = f(\mathbf{x}_j) + \epsilon_j$ of an unknown function $f : \mathcal{X} \mapsto \mathcal{Y}$. Here, $\mathbf{X}^{\mathcal{D}} = \{\mathbf{x}_j\}_{j=1}^m \in \mathcal{X}^m$ denotes training inputs and $\mathbf{y}^{\mathcal{D}} = \{y_j\}_{j=1}^m \in \mathcal{Y}^m$ the corresponding noisy function values. Let $h_\theta : \mathcal{X} \to \mathcal{Y}$ be a function parametrized by a NN with weights $\theta \in \Theta$. For regression, where $\mathcal{Y} \subseteq \mathbb{R}$, we can use $h_\theta$ to define a conditional distribution over the noisy observations $p(y|\mathbf{x}, \theta) = \mathcal{N}(y|h_\theta(\mathbf{x}), \sigma^2)$, with flexible mean represented by $h_\theta$ and observation noise variance $\sigma^2$. Given a prior distribution $p(\theta)$ over the model parameters $\theta$, Bayes' theorem yields a posterior distribution $p(\theta|\mathbf{X}^{\mathcal{D}}, \mathbf{y}^{\mathcal{D}}) \propto p(\mathbf{y}^{\mathcal{D}}|\mathbf{X}^{\mathcal{D}}, \theta)p(\theta)$, where $p(\mathbf{y}^{\mathcal{D}}|\mathbf{X}^{\mathcal{D}}, \theta) = \prod_{j=1}^m p(y_j|\mathbf{x}_j, \theta)$. For an unseen test point $\mathbf{x}^*$, we compute the predictive distribution, which is defined as $p(y^*|\mathbf{x}^*, \mathbf{X}^{\mathcal{D}}, \mathbf{y}^{\mathcal{D}}) = \int p(y^*|\mathbf{x}^*, \theta)p(\theta|\mathbf{X}^{\mathcal{D}}, \mathbf{y}^{\mathcal{D}})d\theta$ and obtained by marginalizing out the posterior over parameters $\theta$.

**BNN inference in the function space.**  Posterior inference for BNNs is difficult due to the high-dimensional parameter space $\Theta$ and the over-parameterized nature of the NN mapping $h_\theta(x)$.

An alternative approach views BNN inference in the function space, i.e., the space of regression functions $h : \mathcal{X} \mapsto \mathcal{Y}$, yielding the Bayes rule $p(h|\mathbf{X}^{\mathcal{D}}, \mathbf{y}^{\mathcal{D}}) \propto p(\mathbf{y}^{\mathcal{D}}|\mathbf{X}^{\mathcal{D}}, h)p(h)$ (Wang et al., 2019; Sun et al., 2019). Here, $p(h)$ is a stochastic process prior with index set $\mathcal{X}$, taking values in $\mathcal{Y}$.

Stochastic processes can be understood as infinite-dimensional random vectors, and, thus, are hard to work with computationally. However, given finite measurement sets $\mathbf{X} := [\mathbf{x}_1, ..., \mathbf{x}_k] \in \mathcal{X}^k, k \in \mathbb{N}$, the stochastic process can be characterized by its corresponding marginal distributions of function values $\rho(\mathbf{h}^{\mathbf{X}}) := \rho(h(\mathbf{x}_1), ...h(\mathbf{x}_k))$ (cf. Kolmogorov Extension Theorem, Oksendal (2013)). Thus, we can break down the functional posterior into a more tractable form by re-phrasing it in terms of posterior marginals based on measurement sets $\mathbf{X}$: $p(\mathbf{h}^{\mathbf{X}}|\mathbf{X}, \mathbf{X}^{\mathcal{D}}, \mathbf{y}^{\mathcal{D}}) \propto p(\mathbf{y}^{\mathcal{D}}|\mathbf{h}^{\mathbf{X}^{\mathcal{D}}})p(\mathbf{h}^{\mathbf{X}})$.

This functional posterior can be tractably approximated to achieve functional BNN inference (Sun et al., 2019; Wang et al., 2019). We briefly describe how this can be done via functional Stein Variational Gradient Descent (fSVGD, Wang et al., 2019). The procedure is explained in more detail in Appendix A.1. The fSVGD algorithm approximates the posterior using a set of $L$ NN parameter particles $\{\theta_1, ..., \theta_L\}$. To optimize posterior approximation, fSVGD iteratively re-samples measurement sets $\mathbf{X}$ from a measurement distribution $\nu$, supported on $\mathcal{X}$, computes SVGD updates (Liu & Wang, 2016) in the function space, and projects them back into the parameter space, in order to update the parameter-space particle configuration approximating the posterior. To achieve this, fSVGD uses the functional posterior score, i.e., the gradient of the log-density $\nabla_{\mathbf{h}^{\mathbf{X}}} \ln p(\mathbf{h}^{\mathbf{X}}|\mathbf{X}, \mathbf{X}^{\mathcal{D}}, \mathbf{y}^{\mathcal{D}}) = \nabla_{\mathbf{h}^{\mathbf{X}}} \ln p(\mathbf{y}^{\mathcal{D}}|\mathbf{h}^{\mathbf{X}^{\mathcal{D}}}) + \nabla_{\mathbf{h}^{\mathbf{X}}} \ln p(\mathbf{h}^{\mathbf{X}})$ to guide the particles towards areas of high posterior probability. Formally, for all $l = 1, ..., L$ the particle updates are computed as

$$\theta^l \leftarrow \theta^l - \gamma \left( \nabla_{\theta^l} \mathbf{h}_{\theta^l}^{\mathbf{X}} \right)^\top \underbrace{\left( \frac{1}{L} \sum_{i=1}^{L} \mathbf{K}_{li} \nabla_{\mathbf{h}_{\theta^i}^{\mathbf{X}}} \ln p(\mathbf{h}_{\theta^l}^{\mathbf{X}}|\mathbf{X}, \mathbf{X}^{\mathcal{D}}, \mathbf{y}^{\mathcal{D}}) + \nabla_{\mathbf{h}_{\theta^l}^{\mathbf{X}}} \mathbf{K}_{li} \right)}_{\text{SVGD update in the function space}} \quad (1)$$

where $\mathbf{K} = [k(\mathbf{h}_{\theta^l}^{\mathbf{X}}, \mathbf{h}_{\theta^i}^{\mathbf{X}})]_{li}$ is the kernel matrix between the function values in the measurement points based on a positive semi-definite kernel function $k(\cdot, \cdot)$. A key insight that will later draw upon in the paper is that such functional approximate inference techniques only use the prior scores and, in principle, do not require a full prior density that integrates to 1.

# 4 META-LEARNING AS SCORE ESTIMATION IN THE FUNCTION SPACE

## 4.1 PROBLEM STATEMENT: META-LEARNING

Meta-learning extracts prior knowledge (i.e., inductive bias) from a set of related learning tasks to accelerate inference on a new and unseen learning task. The meta-learner is given $n$ datasets $\mathcal{D}_1, ..., \mathcal{D}_n$, where each dataset $\mathcal{D}_i = (\mathbf{X}_i^{\mathcal{D}}, \mathbf{y}_i^{\mathcal{D}})$ consists of $m_i$ noisy function evaluations $y_{i,t} = f_i(\mathbf{x}_{i,t}) + \epsilon$ corresponding to a function $f_i : \mathcal{X} \mapsto \mathcal{Y} \subseteq \mathbb{R}$ and additive $\sigma$ sub-Gaussian noise $\epsilon$. In short, we write $\mathbf{X}_i^{\mathcal{D}} = (\mathbf{x}_{i,1}, ..., \mathbf{x}_{i,m_i})^\top$ for the matrix of function inputs and $\mathbf{y}_i^{\mathcal{D}} = (y_{i,1}, ..., y_{i,m_i})^\top$ for the vector of corresponding observations. Following previous work (e.g., Baxter, 2000; Pentina & Lampert, 2014; Rothfuss et al., 2021a), we assume that the functions $f_i \sim \mathcal{T}$ are sampled i.i.d. from a task distribution $\mathcal{T}$, which can be thought of as a stochastic process $p(f)$ that governs the random function $f : \mathcal{X} \mapsto \mathcal{Y}$.

Our goal is to *extract knowledge from the observed datasets, which can then be used as a form of prior for learning a new, unknown target function $f^* \sim \mathcal{T}$ from a corresponding dataset $\mathcal{D}^*$.* By performing Bayesian inference with a meta-learned prior that is attuned to the task distribution $\mathcal{T}$, we hope to obtain posterior predictions that generalize better.

## 4.2 SHORTCOMINGS OF META-LEARNING PRIORS IN THE PARAMETER SPACE

Previous work phrases meta-learning as hierarchical Bayesian problems and tries to meta-learn a prior distribution $p(\theta)$ on the NN weight space that resembles $p(f)$ (e.g., Amit & Meir, 2018; Rothfuss et al., 2021a). This approach suffers from the following central issues: Due to the over-parameterized nature of neural networks, a large set of parameters $\theta$ correspond to exactly the same mapping $h_\theta(\cdot)$. This makes specifying a good prior distribution $p(\theta)$ typically very difficult. At the same time, there is a lack of sufficiently rich parametric families of distributions over the high-dimensional parameter space $\Theta$ that are also computationally viable. So far, only the very restrictive Gaussian family of priors with diagonal covariance matrices has been employed. As a result, the

meta-learned posterior lacks the necessary flexibility to match the complex probabilistic structure of the data-generative process accurately.

### 4.3 Meta-Learning as Score Estimation on the Data-Generating Process

Aiming to address these issues, we acquire prior knowledge in a data-driven manner with a new perspective. We develop a novel approach to meta-learning which hinges upon three key ideas:

First, we view BNN inference in the function space (see Sec. 3), i.e., as posterior inference $p(h|\mathbf{X}^{\mathcal{D}}, \mathbf{y}^{\mathcal{D}}) \propto p(\mathbf{y}^{\mathcal{D}}|\mathbf{X}^{\mathcal{D}}, h)p(h)$ over neural network mappings $h_\theta : \mathcal{X} \mapsto \mathcal{Y}$ instead of parameters $\theta$. From this viewpoint, the prior, which is the target of our meta-learning problem, is $p(h)$, a stochastic process on the function space. This alleviates the aforementioned problem of over-parametrization, which arises when considering priors on the parameter space.

Second, we ask ourselves what is a desirable prior: from a Bayesian perspective, the best possible prior is the stochastic process of the task generating distribution $\mathcal{T}$ itself, i.e., $p(h) = p(f)$. Hence, we would want to meta-learn a prior that matches $p(f)$ as closely as possible. Since the meta-training datasets $\mathcal{D}_1, ..., \mathcal{D}_n$ constitute noisy observations of the function draws $f_1, ...., f_n \sim p(f)$, we can use them to estimate the stochastic process marginals $p(\mathbf{f}^{\mathbf{X}})$. Crucially, we tractably represent and estimate the stochastic process $p(f)$ through its finite marginals $p(\mathbf{f}^{\mathbf{X}})$ in measurement sets $\mathbf{X}$.

Our third insight is that all popular approximate inference methods for BNNs only use the prior score, i.e., the gradient of the log-prior, and not the prior distribution itself. In addition, parametrizing and estimating the prior score is computationally easier than the prior distribution itself since it does not have to integrate to 1, allowing for much more flexible neural network representations. For this reason, our meta-learning approach directly forms an estimate of the scores $s(\mathbf{f}^{\mathbf{X}}, \mathbf{X}) = \nabla_{\mathbf{f}^{\mathbf{X}}} \ln p(\mathbf{f}^{\mathbf{X}})$ of the data-generating process marginals.

In summary, our high-level approach is to *meta-learn / estimate the prior score* $\nabla_{\mathbf{h}^{\mathbf{X}}} \ln p(\mathbf{h}^{\mathbf{X}})$ *that matches data-generating stochastic process from meta-training data* $\mathcal{D}_1, ..., \mathcal{D}_n$. At meta-test time, the meta-learned prior marginals are used in the approximate functional BNN inference on a target dataset $\mathcal{D}^*$ to infuse the acquired prior knowledge into the posterior predictions. In the following section, we discuss in more detail how to implement this general approach as a concrete, computationally feasible algorithm with strong empirical performance.

## 5 The MARS meta-learning algorithm

In Section 4, we have motivated and sketched the idea of meta-learning via score estimation of the data-generating process marginals. We now discuss particular challenges and outline the design choices for translating this idea into a practical meta-learning algorithm.

Score estimation of stochastic process marginals has previously been used in the context of existing functional BNN approaches (Wang et al., 2019; Sun et al., 2019). However, they presume a known stochastic process prior with oracle access to arbitrarily many sampled functions that can be evaluated in arbitrary locations of the domain $\mathcal{X}$. Compared to such a stochastic process prior with oracle access, we face two key challenges which make the score estimation problem much harder:

1. We only have access to a finite number of datasets $\{\mathcal{D}_i\}_{i=1}^n$, each corresponding to one function $f_i$ drawn from the data-generating stochastic process. This means, we have so estimate the marginal scores $\nabla_{\mathbf{h}^{\mathbf{X}}} \ln p(\mathbf{h}^{\mathbf{X}})$ from only $n$ function samples without over-fitting.

2. Per function $f_i$, we only have a limited number of $m_i$ noisy function evaluations $\mathbf{y}_i^{\mathcal{D}}$ in input locations $\mathbf{X}_i^{\mathcal{D}}$ which are given to us exogenously, and we have no control over them. However, to perform score estimation for the marginal $p(\mathbf{h}^{\mathbf{X}})$ distributions, for each function, we need a vector of function values $\mathbf{f}_i^{\mathbf{X}}$ evaluated in the measurements points $\mathbf{X}$.

In the following subsections, we present our approach to estimating the stochastic prior scores and discuss how to solve these challenges. In addition, we discuss how to perform functional approximate BNN inference with the meta-learned score estimates.

### 5.1 Parametric Score Matching for Stochastic Process Marginals

Performing functional approximate inference with fSVGD updates as in (1) requires estimation of the prior marginal scores $\nabla_{\mathbf{h}^{\mathbf{X}}} \ln p(\mathbf{h}^{\mathbf{X}})$ for arbitrary measurement sets $\mathbf{X}$. A key property of

stochastic process marginal scores is their permutation equivariance: if we permute the order of the points in the measurement set, the rows of the score permute in the same manner. Similarly, the measurement sets can be of different sizes, implying score vectors/matrices of different dimensionalities.

Aiming to embed these properties into a parametric model of the prior marginal scores, we use a transformer encoder architecture (Vaswani et al., 2017), that takes as an input a measurement set $\mathbf{X} \in \mathbb{R}^{\dim(\mathcal{X}) \times k}$ of $k$ points and corresponding query function values $\mathbf{h}^{\mathbf{X}} \in \mathbb{R}^{\dim(\mathcal{Y}) \times k}$ and performs attention over the second dimension, i.e., the $k$ columns corresponding to measurement points. Our score network model is denoted as $\mathbf{s}_\phi(\mathbf{h}^{\mathbf{X}}, \mathbf{X})$ with trainable parameters $\phi$ and outputs an estimate of the score, i.e., $\mathbf{s}_\phi(\mathbf{h}^{\mathbf{X}}, \mathbf{X}) \mapsto \tilde{\nabla}_{\mathbf{h}^{\mathbf{X}}} p(\mathbf{h}^{\mathbf{X}})$, a matrix of the same size as the query function values. The model is permutation equivariant w.r.t. the columns corresponding to measurement points and supports inputs/outputs for varying measurement set sizes $k$. We schematically illustrate the score network architecture in Figure 4 and provide details in Appendix A.3.

We train the score network with a modified version of the score matching loss (Hyvärinen & Dayan, 2005; Song et al., 2020). This loss minimizes the Fisher divergence between the data-generating process marginals and our score network, based on samples $\mathbf{f}_1^{\mathbf{X}}, ..., \mathbf{f}_n^{\mathbf{X}}$ from the data-generating process marginals. In contrast to the standard score matching loss, our modified loss takes the dependence of stochastic process marginals on their measurement sets into account. It uses an expectation over randomly sampled measurement sets, similar to the functional approximate inference in Section 3.

$$\mathcal{L}(\phi) := \mathbb{E}_{\mathbf{X}} \left[ \frac{1}{n} \sum_{i=1}^{n} \left( \mathrm{tr}(\nabla_{\mathbf{f}_i^{\mathbf{X}}} \mathbf{s}_\phi(\mathbf{f}_i^{\mathbf{X}}, \mathbf{X})) + \frac{1}{2} ||\mathbf{s}_\phi(\mathbf{f}_i^{\mathbf{X}}, \mathbf{X})||_2^2 \right) \right] \tag{2}$$

As noted previously, our meta-training data only provides noisy function evaluations for a limited, exogenously given, set of input locations per task $f_i$. The input locations $\mathbf{X}_i^{\mathcal{D}}$ per task might even differ in number and be non-overlapping. Thus, we do not have access to the function values $\mathbf{f}_1^{\mathbf{X}}, ..., \mathbf{f}_n^{\mathbf{X}}$ for arbitrary measurement sets $\mathbf{X}$ which we require for our score matching loss in (2). In the following section, we discuss how to resolve this problem.

## 5.2 INTERPOLATING THE DATASETS ACROSS $\mathcal{X}$

In our meta-learning setup (see Section 4.1), we only receive noisy function evaluations $\mathbf{y}_i^{\mathcal{D}}$ in $m_i$ locations $\mathbf{X}_i^{\mathcal{D}}$ for each meta-training task. However, to minimize the score matching loss in (2), we need the corresponding function values in arbitrary measurement locations in the domain $\mathcal{X}$.

Hence, we interpolate each dataset $\mathcal{D}_i$ by a regression model. Importantly, the regression model should be able to quantify the epistemic uncertainty that stems from having observations at only a finite subset of points $\mathbf{X}_i^{\mathcal{D}} \subset \mathcal{X}$. While a range of method could be used, we employ a Gaussian Process (GP) with Matérn$-5/2$ kernel or a Bayesian neural network (BNN) based on Monte Carlo dropout (Gal & Ghahramani, 2016) for this purpose. We fit the respective Bayesian model independently on each dataset $\mathcal{D}_i$ which gives us a posterior $p(\mathbf{f}_i^{\mathbf{X}}|\mathbf{X}, \mathbf{X}_i^{\mathcal{D}}, \mathbf{y}_i^{\mathcal{D}})$ over function values $\mathbf{f}_i^{\mathbf{X}}$ for arbitrary measurement sets $\mathbf{X}$. In Appx. A.4 we provide more details on how we fit each model.

While we could simply use posterior mean values for computing the score matching loss in 2, this would not reflect the epistemic interpolation uncertainty. Instead, for each task, we sample function values $\tilde{\mathbf{f}}_i^{\mathbf{X}} \sim p(\mathbf{f}_i^{\mathbf{X}}|\mathbf{X}, \mathbf{X}_i^{\mathcal{D}}, \mathbf{y}_i^{\mathcal{D}})$ from the corresponding posterior and use them as an input to the score matching loss in (2). Correspondingly, the score matching loss modifies to

$$\tilde{\mathcal{L}}(\phi) := \mathbb{E}_{\mathbf{X}} \left[ \frac{1}{n} \sum_{i=1}^{n} \mathbb{E}_{p(\tilde{\mathbf{f}}_i^{\mathbf{X}}|\mathbf{X}, \mathbf{X}_i^{\mathcal{D}}, \mathbf{y}_i^{\mathcal{D}})} \left[ \mathrm{tr}(\nabla_{\tilde{\mathbf{f}}_i^{\mathbf{X}}} \mathbf{s}_\phi(\tilde{\mathbf{f}}_i^{\mathbf{X}}, \mathbf{X})) + \frac{1}{2} ||\mathbf{s}_\phi(\tilde{\mathbf{f}}_i^{\mathbf{X}}, \mathbf{X})||_2^2 \right] \right] . \tag{3}$$

The further a measurement point $x$ is away from the closest point in $\mathbf{X}_i^{\mathcal{D}}$, the higher is the (epistemic) uncertainty of the corresponding function value $f_i(x)$. By sampling from the posterior, in expectation, the loss in (3) effectively propagates the interpolation uncertainty into the score matching procedure, preventing over-confident score estimates for areas of the domain $\mathcal{X}$ with scarce data. This is illustrated in Appx. D, Figure 6.

## 5.3 PREVENTING META-OVERFITTING OF THE PRIOR SCORE NETWORK

The final challenge we need to address is over-fitting to the meta-training tasks (Yin et al., 2020; Rothfuss et al., 2021a). As we only have meta-training data $\{\mathcal{D}_i\}_{i=1}^n$, corresponding to $n$ functions

---

**Algorithm 1** MARS: Meta-Learning the Data-Generating Process Score

> **Input:** datasets $\mathcal{D}_1, ..., \mathcal{D}_n$, measurement point distribution $\nu$, step size $\eta$
> Initialize score network parameters $\phi$
> **for** $i = 1, ..., n$ **do**
>     fit GP or BNN on $\mathcal{D}_i$, obtain posterior $p(f_i | \mathbf{X}_i^{\mathcal{D}}, \mathbf{y}_i^{\mathcal{D}})$
> **while** not converged **do**
>    $\mathbf{X} \overset{iid}{\sim} \nu$                                     // sample measurement set
>    **for** $i = 1, ..., n$ **do**
>       $\tilde{\mathbf{f}}_i^{\mathbf{X}} \sim p(\mathbf{f}_i^{\mathbf{X}} | \mathbf{X}, \mathbf{X}_i^{\mathcal{D}}, \mathbf{y}_i^{\mathcal{D}})$         // sample function values from posterior marginal
>    $\hat{\mathcal{L}}(\phi) \leftarrow \frac{1}{n} \sum_{i=1}^{n} \left( \text{tr}(\nabla_{\tilde{\mathbf{f}}_i^{\mathbf{X}}} \mathbf{s}_\phi(\tilde{\mathbf{f}}_i^{\mathbf{X}}, \mathbf{X})) + \frac{1}{2} ||\mathbf{s}_\phi(\tilde{\mathbf{f}}_i^{\mathbf{X}}, \mathbf{X})||_2^2 \right)$    // score matching loss
>    $\phi \leftarrow \phi + \eta \nabla_\phi \hat{\mathcal{L}}(\phi)$                  // score network gradient update
> **Output:** trained score network $\mathbf{s}_\phi$

---

$f_i$ drawn from the data-generating stochastic process, i.e., we only have $n$ samples for estimation of marginal scores $\nabla_{\mathbf{h}^{\mathbf{X}}} \ln p(\mathbf{h}^{\mathbf{X}})$, making us prone to overfitting the prior score network. If we performed functional BNN inference with such an overfitted score estimate, the BNN's posterior predictions are likely to be over-confident and too biased towards the functions seen during meta-training.

To counteract the tendency to overfit, we regularize the score network via spectral normalization (Miyato et al., 2018) of the linear layers in the transformer encoder blocks. Spectral normalization controls the Lipschitz constant of the neural network layers by dividing their weight matrix $\mathbf{W}$ by its spectral norm $||\mathbf{W}||$, i.e., re-parametrizing the weights as $\check{\mathbf{W}} := \mathbf{W}/||\mathbf{W}||$. Hence, by applying spectral normalization, we bias our score network towards smoother score estimates corresponding to marginal distributions $p(\mathbf{h}^{\mathbf{X}})$ with higher entropy. Empirically we find that spectral normalization effectively combats meta-overfitting and prevents the estimated prior score from inducing over-confident posterior predictions when employed in functional BNNs. In our experiments, we examined several other regularization methods, e.g., gradient penalties (Gulrajani et al., 2017) or a spectral regularization penalty in the loss, but found spectral normalization to work the best.

### 5.4 THE FULL MARS ALGORITHM

We now summarize our meta-learning algorithm MARS. It consists of two stages:

**Stage 1: Meta-Learning the Prior Score Network.** After initializing the parameters of the score network, we fit a GP or BNN to each of the $n$ meta-training datasets $\mathcal{D}_i$. Then, we train the score network by stochastic gradient descent on the modified score matching loss $\tilde{\mathcal{L}}(\phi)$ in (3). In each iteration, we first sample a measurement set $\mathbf{X}$ i.i.d. from the measurement distribution $\nu = \mathcal{U}(\tilde{\mathcal{X}})$, chosen as uniform distribution over the hypercube $\tilde{\mathcal{X}} \subseteq \mathcal{X}$ conservatively covering the data in $\mathcal{X}$. Based on the measurement set, we sample a vector of functions values $\tilde{\mathbf{f}}_i^{\mathbf{X}}$ from the corresponding GP or BNN posterior marginals $p(\mathbf{f}_i^{\mathbf{X}} | \mathbf{X}, \mathbf{X}_i^{\mathcal{D}}, \mathbf{y}_i^{\mathcal{D}}), i = 1, ..., n$. Based on these samples, we form an empirical estimate $\hat{\mathcal{L}}(\phi)$ of the score matching loss and perform a gradient update step on the score network parameters $\phi$. This is repeated till convergence and summarized in Alg. 1. Depending on whether we use a GP or BNN as interpolator, we refer to our method as MARS-GP or MARS-BNN.

**Stage 2: Functional BNN Inference with the Prior Score Network.** When concerned with a target learning task with a training dataset $\mathcal{D}^* = (\mathbf{X}_*^{\mathcal{D}}, \mathbf{y}_*^{\mathcal{D}})$, we can infuse the meta-learned inductive bias into the BNN inference by using the prior score network $\mathbf{s}_\phi$ from Stage 1 for the approximate inference. In particular, we can either perform functional VI (Sun et al., 2019) or fSVGD (Wang et al., 2019), using the score network $\mathbf{s}_\phi(\mathbf{h}^{\mathbf{X}}, \mathbf{X})$ predictions as swap-in replacement for the marginal scores $\nabla_{\mathbf{h}^{\mathbf{X}}} \ln p(\mathbf{h}^{\mathbf{X}})$ of a user-specified stochastic process prior. In our experiments, we use fSVGD (see Section 3) as functional approximate inference method. The resulting fSVGD BNN inference procedure with our meta-learned priors score network is summarized in Alg. 2 in Appx. A.

## 6 EXPERIMENTS

We provide a detailed benchmark comparison with existing meta-learning methods, demonstrating that MARS: *(i)* achieves state-of-the-art performance in terms of predictive accuracy, *(ii)* yields

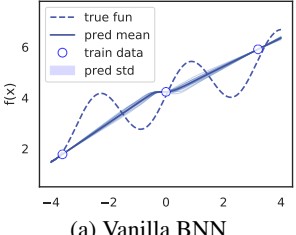
(a) Vanilla BNN

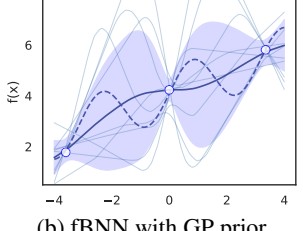
(b) fBNN with GP prior

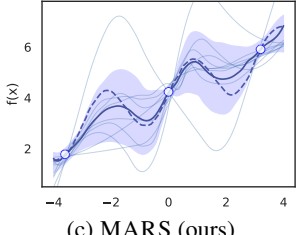
(c) MARS (ours)

Figure 1: BNN posterior predictions with (a) zero-centered Gaussian prior on the NN parameters $\theta$ (b) Gaussian process prior and (c) meta-learned MARS prior scores.

| | SwissFEL | Physionet-GCS | Physionet-HCT | Berkeley-Sensor | Argus-Control |
|---|---|---|---|---|---|
| Vanilla GP | $0.876 \pm 0.000$ | $2.240 \pm 0.000$ | $2.768 \pm 0.000$ | $0.258 \pm 0.000$ | $0.026 \pm 0.000$ |
| Vanilla BNN | $0.529 \pm 0.022$ | $2.664 \pm 0.274$ | $3.938 \pm 0.869$ | $0.151 \pm 0.018$ | $0.016 \pm 0.002$ |
| MAML | $0.730 \pm 0.057$ | $1.895 \pm 0.141$ | $2.413 \pm 0.113$ | $\mathbf{0.121 \pm 0.027}$ | $0.017 \pm 0.001$ |
| BMAML | $0.577 \pm 0.044$ | $1.894 \pm 0.062$ | $2.500 \pm 0.002$ | $0.222 \pm 0.032$ | $0.037 \pm 0.003$ |
| NP | $0.471 \pm 0.053$ | $2.056 \pm 0.209$ | $2.594 \pm 0.107$ | $0.173 \pm 0.018$ | $0.020 \pm 0.001$ |
| PACOH-GP | $\mathbf{0.376 \pm 0.024}$ | $\mathbf{1.498 \pm 0.081}$ | $2.361 \pm 0.047$ | $0.197 \pm 0.058$ | $0.016 \pm 0.005$ |
| PACOH-NN | $0.437 \pm 0.021$ | $1.623 \pm 0.057$ | $2.405 \pm 0.017$ | $0.160 \pm 0.070$ | $0.018 \pm 0.002$ |
| MARS-GP (ours) | $\mathbf{0.391 \pm 0.011}$ | $\mathbf{1.471 \pm 0.083}$ | $\mathbf{2.309 \pm 0.041}$ | $\mathbf{0.116 \pm 0.024}$ | $\mathbf{0.013 \pm 0.001}$ |
| MARS-BNN (ours) | $0.407 \pm 0.061$ | $\mathbf{1.307 \pm 0.065}$ | $2.248 \pm 0.057$ | $\mathbf{0.113 \pm 0.015}$ | $0.017 \pm 0.003$ |

Table 1: Meta-Learning benchmark results in terms of the test RMSE. Reported are the mean and standard deviation across five seeds. MARS consistently yields the most accurate predictions.

well-calibrated uncertainty estimates. In a comprehensive ablation study, we shed more light on the central components of MARS and provide empirical support for our design decisions.

## 6.1 EXPERIMENT SETUP

**Environments.** We consider five realistic meta-learning environments for *regression*. The first environment corresponds to data of different calibration sessions of the Swiss Free Electron Laser (*SwissFEL*) (Milne et al., 2017). Here, a task requires predicting the laser's beam intensity based on undulators parameters. We also consider electronic health measurements (*PhysioNet*) from intensive care patients (Silva et al., 2012), in particular, the Glasgow Coma Scale (*GCS*) and the hematocrit value (*HCT*). Here, each task correspond to a patient. We also use the Intel Berkeley Lab temperature sensor dataset (*Berkeley-Sensor*) (Madden, 2004), requiring auto-regressive prediction of temperature measurements of sensors in different locations of the building. Finally, the *Argus-Control* environment requires predicting the total variation of a robot from its target position based on its PID controller parameters (Rothfuss et al., 2022). Here, tasks correspond to different step sizes between the source and target location. See Appendix B for details.

**Baselines.** As non-meta-learning baselines, we use a *Vanilla GP* with RBF kernel and a *Vanilla BNN* with a zero-mean, spherical Gaussian prior and SVGD inference (Liu & Wang, 2016). To compare to previous meta-learners, we report results for model agnostic meta-learning (*MAML*) (Finn et al., 2017), Bayesian MAML (*BMAML*) (Yoon et al., 2018) and neural processes (*NPs*) (Garnelo et al., 2018). In addition, we include *PACOH* (Rothfuss et al., 2021a) which performs hierarchical Bayes inference to meta-learn a GP prior (*PACOH-GP*) or prior over NN parameters (*PACOH-NN*).

## 6.2 EMPIRICAL BENCHMARK STUDY

**Qualitative illustration.** Fig. 1 illustrates the posterior predictions of a) a BNN with Gaussian Prior in the parameter space, b) a functional BNN with GP prior (Wang et al., 2019) and c) a fBNN trained with the meta-learned MARS-GP marginal scores. For the meta-training we use $n = 20$ tasks with each $m = 8$ data points, generated from a student-t process with sinusoidal mean function $2x + 5\sin(2x)$. Similar to previous work (e.g., Fortuin et al., 2022) we observe that weight-space BNN priors provide poor inductive bias and yield grossly over-confident uncertainty estimates. Approaching the BNN inference problem in the function space (see Fig 1b) results in much better uncertainty estimates. In this case, however, the uninformative GP prior does not result in accurate mean predictions. Finally, the meta-learned scores in the function space provide useful inductive biased towards the linear+sinusoidal data-generating pattern while yielding tight but reliable confidence intervals.

| | SwissFEL | Physionet-GCS | Physionet-HCT | Berkeley-Sensor | Argus-Control |
|---|---|---|---|---|---|
| Vanilla GP | $0.135 \pm 0.000$ | $0.268 \pm 0.000$ | $0.277 \pm 0.000$ | $0.119 \pm 0.000$ | $0.090 \pm 0.000$ |
| Vanilla BNN | $0.085 \pm 0.008$ | $0.277 \pm 0.013$ | $0.307 \pm 0.009$ | $0.206 \pm 0.025$ | $0.104 \pm 0.005$ |
| BMAML | $0.115 \pm 0.036$ | $0.279 \pm 0.010$ | $0.423 \pm 0.106$ | $0.154 \pm 0.021$ | $0.068 \pm 0.005$ |
| NP | $0.131 \pm 0.056$ | $0.299 \pm 0.012$ | $0.319 \pm 0.004$ | $0.140 \pm 0.035$ | $0.094 \pm 0.015$ |
| PACOH-GP | $\mathbf{0.038 \pm 0.006}$ | $\mathbf{0.262 \pm 0.004}$ | $0.296 \pm 0.003$ | $0.251 \pm 0.035$ | $0.102 \pm 0.010$ |
| PACOH-NN | $\mathbf{0.037 \pm 0.005}$ | $0.267 \pm 0.005$ | $0.302 \pm 0.003$ | $0.223 \pm 0.012$ | $0.119 \pm 0.005$ |
| MARS-GP (ours) | $\mathbf{0.035 \pm 0.002}$ | $\mathbf{0.263 \pm 0.001}$ | $\mathbf{0.136 \pm 0.007}$ | $\mathbf{0.080 \pm 0.005}$ | $\mathbf{0.055 \pm 0.002}$ |
| MARS-BNN (ours) | $0.054 \pm 0.009$ | $\mathbf{0.268 \pm 0.023}$ | $0.231 \pm 0.029$ | $\mathbf{0.078 \pm 0.020}$ | $0.076 \pm 0.031$ |

Table 2: Meta-learning benchmark results in terms of the calibration error. Reported are the mean and standard deviation across five seeds. MARS provides the best-calibrated uncertainty estimates.

| Estimator | SwissFEL | Physionet-GCS | Physionet-HCT | Berkeley-Sensor | Argus-Control |
|---|---|---|---|---|---|
| MARS-GP | $\mathbf{0.391 \pm 0.011}$ | $\mathbf{1.471 \pm 0.083}$ | $\mathbf{2.309 \pm 0.041}$ | $\mathbf{0.116 \pm 0.024}$ | $\mathbf{0.013 \pm 0.001}$ |
| SSGE score estimates | $0.449 \pm 0.027$ | $3.292 \pm 0.562$ | $2.784 \pm 0.257$ | $1.105 \pm 0.562$ | $0.030 \pm 0.003$ |
| No spectral reg. | $0.420 \pm 0.060$ | $2.208 \pm 0.338$ | $\mathbf{2.560 \pm 0.341}$ | $1.734 \pm 0.169$ | $\mathbf{0.014 \pm 0.001}$ |
| No GP sampling | $0.471 \pm 0.059$ | $2.994 \pm 0.363$ | $5.995 \pm 1.108$ | $1.253 \pm 0.112$ | $0.073 \pm 0.003$ |

Table 3: Ablation study results for MARS components in terms of the RMSE.

**MARS provides accurate predictions.** We perform a comprehensive benchmark study with the environments and baselines introduced in Sec. 6.1. Table 1 reports RMSE on unseen meta-test tasks. Both MARS-GP and MARS-BNN yield substantial improvements in the RMSE compared to the non-meta-learning baselines and significantly outperforms all other meta-learning baselines in most environments. This shows that MARS can acquire valuable inductive biases from data reliably. We hypothesize that MARS performs substantially better than PACOH-NN, which meta-learns restrictive diagonal Gaussian priors over $\Theta$ since it has much more flexibility in expressing prior knowledge about the data-generating process. In the majority of environments, MARS also outperforms PACOH-GP which meta-learns a GP prior. In contrast to Gaussian marginals, MARS can meta-learn any stochastic process marginal, and, thus has much more flexibility to express inductive bias.

**MARS yields well-calibrated uncertainty estimates.** We hypothesize that by performing meta-learning in the function space, we avoid the pitfalls of NN over-parametrization, which often lead to over-confidence. To investigate the quality of uncertainty estimates, we compute the *calibration error*, which measures how much the predicted confidence regions deviate from the actual frequencies of test data in the respective regions. We list the results in Table 2[1]. While, in most cases, still better than the baselines, the uncertainty estimates MARS-BNN are on average worse than those of MARS-GP. This is not surprising as the MC-dropout uncertainty estimates are typically not as reliable as those of a GP. MARS-GP consistently yields the best-calibrated uncertainty estimates and, for some environments, reduces the calibration error by more than 30 % compared to the next best baseline. This provides further evidence for the soundness and efficacy of our function space approach to meta-learning. Finally, the superior calibration performance suggests that the spectral regularization, together with the posterior uncertainty sampling in Sec. 5, effectively prevents meta-overfitting and accounts for epistemic uncertainty.

### 6.3 Ablation Study

We empirically investigate the algorithm components introduced in Sec. 5 and provide supporting empirical evidence for our design decisions. First, we perform a quantitative ablation study where we vary/remove components of our algorithm. We consider MARS-GP with 1) nonparametric score estimator SSGE (Shi et al., 2018) instead of the parametric score network + score matching from Sec. 5.1, 2) without spectral regularization, and 3) without GP posterior sampling, i.e., using the GP posterior mean instead of samples for the score matching. Table 3 and 4 report the quantitative results of this ablation experiment. In the following, we discuss the three aspects separately:

**Parametric score matching outperforms nonparametric score estimation.** In Sec. 5.1 we have introduced a parametric score network which we train with the score matching loss. Alternatively, for each measurement set $\mathbf{X}$, we could apply the nonparametric score estimator SSGE (Shi et al., 2018) to the function values $\tilde{\mathbf{f}}_1^{\mathbf{X}}, ..., \tilde{\mathbf{f}}_n^{\mathbf{X}}$, sampled from the posteriors, and directly use the resulting

---

[1]Note that we omit MAML since it does not provide uncertainty estimates.

| Estimator | SwissFEL | Physionet-GCS | Physionet-HCT | Berkeley-Sensor | Argus-Control |
|---|---|---|---|---|---|
| MARS-GP | **0.035 ± 0.002** | 0.263 ± 0.001 | **0.136 ± 0.007** | **0.080 ± 0.005** | **0.055 ± 0.002** |
| SSGE score estimates | 0.151 ± 0.001 | 0.249 ± 0.002 | 0.246 ± 0.007 | 0.232 ± 0.010 | 0.210 ± 0.011 |
| No spectral reg. | 0.233 ± 0.041 | 0.265 ± 0.012 | 0.244 ± 0.009 | 0.192 ± 0.018 | 0.187 ± 0.028 |
| No GP sampling | 0.204 ± 0.013 | **0.225 ± 0.021** | 0.237 ± 0.018 | 0.141 ± 0.029 | 0.216 ± 0.066 |

Table 4: Ablation study results for MARS components in terms of the calibration error.

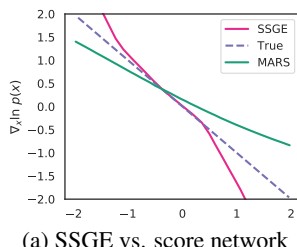
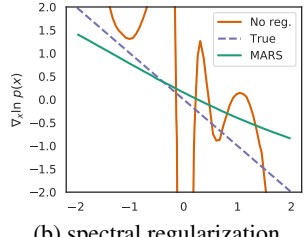
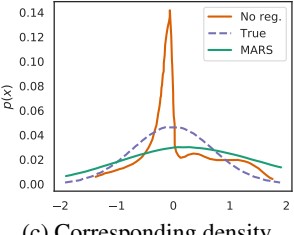

(a) SSGE vs. score network     (b) spectral regularization     (c) Corresponding density

Figure 2: (a) Underconfident MARS and overconfident SSGE score predictions on ten samples from a Gaussian. (b) Score estimates of MARS with and without regularization on ten function samples from a zero mean GP with SE kernel. (c) Numerically integrated score predictions corresponding to the scores in (b). Overall, SSGE and MARS w/o regularization overfit and underestimate the variance, whereas spectral normalization biases MARS towards higher entropy.

score estimates during the fSVGD posterior inference in (1). This produces score estimates 'ad-hoc' and does not require us to train an explicit score network. In contrast, our score network is a global function that explicitly considers the measurement set and thus can exploit similarity structure across $\mathcal{X}$, which SSGE cannot. To SSGE we also cannot simply add inductive bias towards higher entropy as we do through spectral normalization. Overall, the experiment results in Table 3 and 4 suggest that instantiating our general approach with SSGE performs worse than MARS. We can also observe this visually in Fig. 2a, where we plot the score estimate of MARS and SSGE for a GP marginal where the true score is known. While the MARS score network slightly overestimates the variance of true generative-process marginal, SSGE implicitly underestimates the prior variance, leading to over-confident predictions. Finally, Table 7 in Appx. D quantitatively shows that the MARS score network provides the most accurate estimates compared to a variety of nonparametric estimators.

**Spectral normalization prevents meta-overfitting.** In Sec. 5.3, we added spectral normalization to our score network. Here, we empirically investigate what happens when we remove the spectral normalization from MARS. Figure 2b and 2c illustrate how our score network over-fits and under-estimates the true data-generating variance when we do not use spectral regularization. Quantitatively, we observe substantial increases in calibration errors and a consistent worsening of the predictive accuracy. Overall, this highlights the empirical importance of spectral normalization for preventing meta-overfitting and biasing the score estimates toward higher entropy.

**Accounting for epistemic uncertainty of the GP interpolators is crucial.** In MARS, we interpolate each dataset with a GP or BNN and, when performing score estimation, use samples from the posterior marginals to account for the epistemic uncertainty of the interpolation. Here, we empirically study what happens if we ignore the epistemic uncertainty and take each GP's mean predictions. When doing so, we observe detrimental effects on the RMSE and calibration error in the majority of the environments in Table 3 and 4. This affirms that propagating the epistemic uncertainty of the interpolation into the score estimates is a critical component of MARS.

## 7 CONCLUSION

We have introduced a novel meta-learning approach in the function space that estimates the score of the data-generating process marginals from a set of related datasets. When facing a new learning task, we use the meta-learned score network as a source of prior for functional approximate BNN inference. By representing inductive bias as the score of a stochastic process, our approach is versatile and can seamlessly acquire/represent complex prior knowledge. Empirically, this translates into strong performance when compared to previous meta-learning methods. The substantial improvements of MARS in terms of the quality of uncertainty estimates open up many potential extensions toward interactive machine learning where exploration based on epistemic uncertainty is vital.

ACKNOWLEDGMENTS

This research was supported by the European Research Council (ERC) under the European Union's Horizon 2020 research and innovation program grant agreement no. 815943 and the Swiss National Science Foundation under NCCR Automation, grant agreement 51NF40 180545. Jonas Rothfuss was supported by an Apple Scholars in AI/ML fellowship. We thank Alex Hägele, Parnian Kassraie, Lars Lorch and Danica J. Sutherland for their valuable feedback.

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

# A MARS IMPLEMENTATION DETAILS

This section focuses on a detailed explanation of the proposed framework. We first elaborate on the fSVGD inference process and then provide information regarding the network architecture and the hyperparameter configuration.

## A.1 FSVGD INFERENCE

To perform approximate BNN inference with the meta-learned score network, we use *functional Stein Variational Gradient Descent (fSVGD)* (Wang et al., 2018). In the following, we explain our fSVGD implementation and how it interplays with the score network in our MARS framework.

In our context, the goal of fSVGD is to approximate the posterior $p(h_\theta|\mathbf{X}^\mathcal{D}, \mathbf{y}^\mathcal{D}) \propto p(\mathbf{y}^\mathcal{D}|\mathbf{X}^\mathcal{D}, h_\theta)p(h_\theta)$ over neural network mappings $h_\theta$. Recall from Section 3, that $\mathbf{X}^\mathcal{D} = \{\mathbf{x}_j\}_{j=1}^m \in \mathcal{X}^m$ are the training inputs, $\mathbf{y}^\mathcal{D} = \{y_j\}_{j=1}^m \in \mathcal{Y}^m$ the corresponding noisy function values for a new, unseen learning tasks. The fSVGD algorithm approximates the posterior using a set of $L$ NN parameter particles $\{\theta_1, ..., \theta_L\}$ where each $\theta_l$ corresponds to the weights and biases of a neural network. The particles (i.e., weights and biases) are initialized based on the scheme of Steinwart (2019) (see Appendix A.2 for details).

To make the BNN inference in the function space tractable, in each iteration, fSVGD samples a measurement set $\mathbf{X}$ from a measurement distribution $\nu$ and performs its particle updates based on the posterior marginals $p(\mathbf{h}^\mathbf{X}|\mathbf{X}, \mathbf{X}^\mathcal{D}, \mathbf{y}^\mathcal{D}) \propto p(\mathbf{y}^\mathcal{D}|\mathbf{h}^{\mathbf{X}^\mathcal{D}})p(\mathbf{h}^\mathbf{X})$ corresponding to $\mathbf{X}$. We use a uniform distribution over the bounded domain $\mathcal{X}$ as measurement distribution $\nu$ and sample the $L$ measurement points i.i.d. from it.

For each NN particle, we compute the NN function values in the measurement points, i.e. $\mathbf{h}_{\theta^l}^\mathbf{X} = (h_\theta^l(\mathbf{x}_1), ..., h_\theta^l(\mathbf{x}_k))$, $l = 1, ..., L$ and the corresponding posterior marginal scores:

$$\nabla_{\mathbf{h}_{\theta^l}^\mathbf{X}} \ln p(\mathbf{h}_{\theta^l}^\mathbf{X}|\mathbf{X}, \mathbf{y}^\mathcal{D}) \leftarrow \underbrace{\nabla_{\mathbf{h}_{\theta^l}^\mathbf{X}} \ln p(\mathbf{y}^\mathcal{D}|\mathbf{h}_{\theta^l}^{\mathbf{X}^\mathcal{D}})}_{\text{likelihood score}} + \underbrace{\nabla_{\mathbf{h}_{\theta^l}^\mathbf{X}} \ln \hat{p}(\mathbf{h}_{\theta^l}^\mathbf{X})}_{:= \mathbf{s}_\phi(\mathbf{h}_{\theta^l}^\mathbf{X}, \mathbf{X})} \tag{4}$$

Here, we use a Gaussian likelihood $p(\mathbf{y}^\mathcal{D}|\mathbf{h}^{\mathbf{X}^\mathcal{D}}) = \prod_{j=1}^k \mathcal{N}(y_i^\mathcal{D}; h_\theta^l(x_j), \sigma^2)$ where $\sigma^2$ is the likelihood variance. Unlike in Wang et al. (2018) where the stochastic process prior is exogenously given, and its marginals $p(\mathbf{h}^\mathbf{X})$ are approximated as multivariate Gaussians, we use our meta-learned marginal prior scores. In particular, we use the score networks $\mathbf{s}_\phi(\mathbf{h}_{\theta^l}^\mathbf{X}, \mathbf{X})$ output as a swap-in for the prior score $\nabla_{\mathbf{h}_{\theta^l}^\mathbf{X}} \ln \hat{p}(\mathbf{h}_{\theta^l}^\mathbf{X})$. Finally, based on the score in (4) and the function values $\mathbf{h}_{\theta^l}^\mathbf{X}$, we can compute the SVGD updates in the function space, and project them back into the parameter space via the NN Jacobian $\nabla_{\theta^l} \mathbf{h}_{\theta^l}^\mathbf{X}$:

$$\theta^l \leftarrow \theta^l - \gamma \underbrace{(\nabla_{\theta^l} \mathbf{h}_{\theta^l}^\mathbf{X})^\top}_{\text{NN Jacobian}} \underbrace{\left( \frac{1}{L} \sum_{i=1}^L \mathbf{K}_{li} \nabla_{\mathbf{h}_{\theta^i}^\mathbf{X}} \ln p(\mathbf{h}_{\theta^l}^\mathbf{X}|\mathbf{X}, \mathbf{X}^\mathcal{D}, \mathbf{y}^\mathcal{D}) + \nabla_{\mathbf{h}_{\theta^l}^\mathbf{X}} \mathbf{K}_{li} \right)}_{\text{SVGD update in the function space}} . \tag{5}$$

Here, $\gamma$ is the SVGD step size and $\mathbf{K} = [k(\mathbf{h}_{\theta^l}^\mathbf{X}, \mathbf{h}_{\theta^i}^\mathbf{X})]_{li}$ is the kernel matrix between the function values in the measurement points based on a kernel function $k(\cdot, \cdot) : \mathcal{Y}^k \times \mathcal{Y}^k \mapsto \mathbb{R}$. We use the RBF kernel $k(\mathbf{h}, \mathbf{h}') = \exp\left((-||\mathbf{h} - \mathbf{h}'||^2)/(2\ell_k)\right)$ where $\ell_k$ is the bandwidth hyper-parameter. The particle update in (5) completes one iteration of fSVGD. We list the full procedure in Algorithm 2.

**fSVGD hyperparameter selection.** For the fBNN training using fSVGD, among other parameters, we need to choose the step size $\gamma$, kernel bandwidth $\ell_k$, and likelihood standard deviation $\sigma$, as we found these three to have the most substantial impact on training dynamics. We fix the number of particles to $L = 10$ and perform 10000 fSVGD update steps. We always standardize both the input and output data based on the meta-training data's empirical mean and standard deviation. For the NNs, we use three hidden layers, each of size 32 and with leaky ReLU activations. To initialize the NN weights, we use He initialization with a uniform distribution (He et al., 2015) and the bias initializer of Steinwart (2019) (see Section A.2 for details). Generally, we choose the

**Algorithm 2** Approximate BNN Inference with fSVGD (Wang et al., 2019)

---

**Input:** SVGD kernel function $k(\cdot, \cdot; \ell_k)$, bandwidth $\ell_k$, step size $\gamma$
**Input:** dataset $\mathcal{D}^* = (\mathbf{X}_*^{\mathcal{D}}, \mathbf{y}_*^{\mathcal{D}})$ for target task, trained score network $\mathbf{s}_\phi(\cdot, \cdot)$
Initialize set of BNN particles $\{\theta_1, ..., \theta_L\}$
**while** not converged **do**

    $\mathbf{X} \overset{iid}{\sim} \nu$                                                               // sample measurement set
    **for** $l = 1, ..., L$ **do**
        $\mathbf{h}_{\theta^l}^{\mathbf{X}} \leftarrow (h_\theta^l(\mathbf{x}_1), ..., h_\theta^l(\mathbf{x}_k))$ where $\mathbf{X} = (\mathbf{x}_1, ..., \mathbf{x}_k)$      // compute NN function values in $\mathbf{X}$
        $\nabla_{\mathbf{h}_{\theta^l}^{\mathbf{x}}} \ln p(\mathbf{h}_{\theta^l}^{\mathbf{X}} | \mathbf{X}, \mathbf{y}^{\mathcal{D}}) \leftarrow \nabla_{\mathbf{h}_{\theta^l}^{\mathbf{x}}} \ln p(\mathbf{y}^{\mathcal{D}} | \mathbf{h}_{\theta^l}^{\mathbf{X}^{\mathcal{D}}}) + \mathbf{s}_\phi(\mathbf{h}_{\theta^l}^{\mathbf{X}}, \mathbf{X})$      // posterior marginal score

        $\theta^l \leftarrow \theta^l - \gamma \left( \nabla_{\theta^l} \mathbf{h}_{\theta^l}^{\mathbf{X}} \right)^\top \left( \frac{1}{L} \sum_{i=1}^{L} \mathbf{K}_{li} \nabla_{\mathbf{h}_{\theta^i}^{\mathbf{x}}} \ln p(\mathbf{h}_{\theta^l}^{\mathbf{X}} | \mathbf{X}, \mathbf{y}^{\mathcal{D}}) + \nabla_{\mathbf{h}_{\theta^l}^{\mathbf{x}}} \mathbf{K}_{li} \right)$ // fSVGD update
**Output:** Set of BNN particles $\{\theta_1, ..., \theta_L\}$ that approximate the BNN posterior process

---

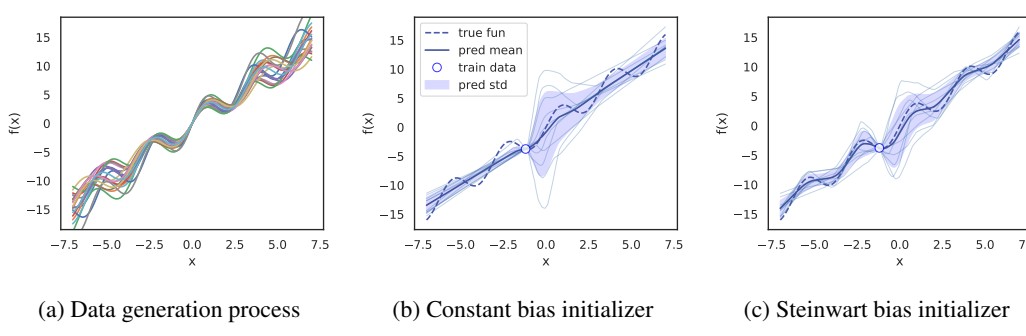

    (a) Data generation process         (b) Constant bias initializer         (c) Steinwart bias initializer

Figure 3: a) The underlying data generating process and corresponding fBNN prediction after training using the fSVGD algorithm for 2000 iterations with b) constant bias initializer, and c) Steinwart bias initializer. With the Steinwart bias initializer, we get the desirable non-linear behavior of the fSVGD BNN much faster.

kernel bandwidth $\ell_k$ via a random hyper-parameter search over the values range of $[0.1, 10]$. When comparing to the original fSVGD implementation with SSGE, we fix the SSGE lengthscale to $0.2$ for comparability. Note that we also experimented with using the median heuristic (as proposed by Shi et al. (2018)); however, we observed this to yield inferior performance.

## A.2 BIAS INITIALIZATION

When initializing the biases to zeros or small positive constants, we find that the learned neural network maps behave like linear functions further away from zero and that it takes many SVGD iterations for them to assume non-linear behavior at the boundaries of the domain.

To address this issue, we use the bias initialization scheme of Steinwart (2019), which initializes the biases in such a way that the kinks of the leaky ReLU functions are more evenly distributed across the domain and less concentrated around zero. More specifically, we initialize each bias $b_i$ as

$$b_i := - \langle w_i, x_i^\star \rangle,$$

where $w_i$ is uniformly sampled from a sphere by taking $w_i = \frac{a_i}{||a_i||}$, $a_i \sim U(0, 1)$. $x_i^*$'s are sampled uniformly: $x_i^* \sim U(min_x, max_x)$, where $min_x$ and $max_x$ are the minimum and maximum points in the input domain respectively.

By using *Steinwart* initialization, the neural networks show much more non-linear behavior after initialization. Compared to constant bias initialization, learning non-linear functional relationships happens much more quickly. We showcase this in Figure 3: Figure 3a displays a simple data generation process of sinusoids of varying amplitude, frequency, phase shift, and slope. Figure 3b displays corresponding BNN predictions after 2000 iterations with a constant bias initializer (where the constant equals 0.01), and Figure 3c shows the result of the same training dynamics, using *Steinwart's* bias initializer instead of the constant initializer. The BNNs with the *Steinwart* initialization

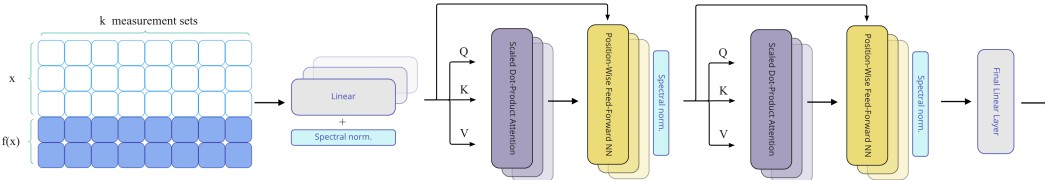

Figure 4: Architecture of MARS score estimation network. From left to right: $k$ measurement sets consisting of input-output concatenations with $x_i \in \mathbb{R}^3, f(x_i) \in \mathbb{R}^2, i = 1, ..., k$; inputs/outputs embeddings using spectrally normalized linear layers; two identical blocks of scaled dot product attention, residual layers and feed-forward position-wise NN with spectrally normalized linear layers; the final linear layer, *not* spectrally normalized.

assume desirable non-linear behavior much earlier during training, thus, significantly speeding up training. However, if trained for a large enough number of iterations, the performance of the two networks with different bias initializations becomes much less distinct. For further details on the implications of the *Steinwart* initializer on the training dynamics, we refer to Steinwart (2019).

### A.3 SCORE ESTIMATION NETWORK

We now give an overview of the score estimation network. We start by providing motivation for the required architecture, detailing the permutation equivariance properties of the network. We construct the proposed architecture step-by-step, giving the architectural details and mentioning additional architectural designs we experimented with. Finally, we comment on the optimization method and acknowledge the libraries we used in our implementations.

**Incorporating task invariances.** The proposed network is permutation equivariant across the $k$ dimension: reordering the measurement inputs would result in reordering the network's prediction in the same manner: Formally, for any permutation $\pi$ of the measurement set indices we have that $\mathbf{s}_\phi(\mathbf{h}^{\mathbf{X}_{\pi(1:d)}}, \mathbf{X}_{\pi(1:d)})_{i,j} \mapsto \tilde{\nabla}_{\mathbf{h}} \mathbf{x}_{1:d} p(\mathbf{h}^{\mathbf{X}_{1:d}})_{\pi(i),\pi(j)}$. To impose permutation equivariance, we use the self-attention mechanism (Vaswani et al., 2017). Permutation equivariance is obtained by concatenating inputs (measurement points) and the corresponding functional evaluations (i.e., the concatenation constructs an object corresponding to Transformer *tokens*), which are then embedded and inputted to the attention mechanism. The architecture is displayed in Figure 4.

**Constructing the network architecture.** The core of our model is composed of *two* identical blocks, each consisting of *two* residual layers, the first one applying multi-head self-attention and the second one position-wise feed-forward neural network, similar to the vanilla Transformer encoder (Vaswani et al., 2017). Since the multi-head attention is permutation equivariant over the measurement point (i.e., token) dimension, the representation is permutation equivariant at all times (Lee et al., 2019). Finally, to minimize training time, we select attention embedding dimensions proportional to the data dimension of the environments; higher-dimensional environments (e.g., SwissFEL, detailed in Appendix B) correspond to the higher number of attention parameters/embedding dimensions and lower environments (e.g., the Sinusoid and Berkeley environment) to lower embeddings. Furthermore, we tune the step size and report the chosen configurations under https://tinyurl.com/376wp8xe. We describe the corresponding implementation details in the following subsection.

**Constructing the score network.** We train the score network on the input/output pair concatenations, which are then embedded onto a higher-dimensional space. After this, we perform self-attention. Specifically, we fix the initialization scale of self-attention weights to 2.0. As mentioned, the size of the model and embeddings varies across meta-learning environments. Next follows a position-wise feed-forward neural network, for which we use Exponential Linear-Unit (ELU) as the activation, as implemented in Haiku's vanilla Transformer encoder (Hennigan et al., 2020). Afterward, we apply the self-attention mechanism again, following a position-wise feed-forward neural network, after which we apply the final linear layer.

**Variants of the attention mechanism.** We experimented with multiple variations of the attention mechanism, all being permutation equivariant in the $k$ dimension, similar to the work of Lorch et al. (2022). We mention two other architectural designs: in the first design, rather than using the embeddings of the concatenation of input-output points as keys, queries, and values (as performed in the vanilla Transformer encoder), we experimented with using the embeddings of the inputs as the keys and queries and the embeddings of functional outputs. However, this design yields varying performance: on several low-dimensional tasks, the performance was slightly better, whereas performance on tasks with higher-dimensional inputs was substantially worse. In another attempt, we experimented with learning different embeddings for inputs and functional outputs, which we then concatenated and used as keys, queries, and values. The difference in performance in this method was marginal, and we decided against it for simplicity.

**Optimizer setup and employed libraries.** Finally, across all experiments, we use gradient clipping of the prior score together with the ADAM optimizer (Kingma & Ba, 2014), with the default values set in Jax (Bradbury et al., 2018) and Haiku (Hennigan et al., 2020). For the gradient clipping, we use values of 1., 10., or 100., depending on the task and the underlying properties of the data generating process. For example, for the first experiment in Appendix D, larger clipping values (50 or 100) performed better for the heavier-tailed Student's-$t$ process, whereas clipping at 10. resulted in a good performance of the GP task. We train the score network for 20000 iterations. For the GP and Student-$t$ process implementations, we use GP-Jax (Pinder & Dodd, 2022), scikit-learn (Pedregosa et al., 2011) and TensorFlow Distributions packages (Dillon et al., 2017).

## A.4 INTERPOLATING THE DATASETS ACROSS $\mathcal{X}$

**Interpolation with Gaussian Processes.** In GP regression, each data point corresponds to a feature-target tuple $z_{i,j} = (\mathbf{x}_{i,j}, y_{i,j}) \in \mathbb{R}^d \times \mathbb{R}$. For the $i$-th dataset, we write $\mathcal{D}_i = (\mathbf{X}_i, \mathbf{y}_i)$, where $\mathbf{X}_i = (x_{i,1}, \ldots, x_{i,m_i})^\top$ and $\mathbf{y}_i = (y_{i,1}, \ldots, y_{i,m_i})^\top$. GPs are a Bayesian method in which the prior $\mathcal{P}(h) = \mathcal{GP}\left(h \mid m(x), k\left(x, x'\right)\right)$ is specified by a positive definite kernel $k : \mathcal{X} \times \mathcal{X} \to \mathbb{R}$ and a mean function $m : \mathcal{X} \to \mathbb{R}$. In this section, we assume a zero mean GP and omit writing the dependence on $m(\cdot)$. As the GP kernel, we use the Matérn covariance function

$$k(\mathbf{x}, \mathbf{x}'; \ell, \nu) = \frac{2^{1-\nu}}{\Gamma(\nu)} \left(\sqrt{2\nu} \left\|\frac{\mathbf{x} - \mathbf{x}'}{\ell}\right\|_2\right)^\nu \mathrm{K}_\nu \left(\sqrt{2\nu} \left\|\frac{\mathbf{x} - \mathbf{x}'}{\ell}\right\|_2\right), \tag{6}$$

with degree $\nu$, lengthscale $\ell$, and $\Gamma(\cdot)$ representing the gamma function, and $K_v$ the modified Bessel function of the second kind. We use fix the degree of the Matérn kernel to $\nu = 5/2$ and choose the lengthscale $\ell$ via 4-fold cross-validation (CV) from 10 log-uniformly spaced points in $[0.001, 10]$. We select the lengthscale that maximizes the 4-fold CV log marginal likelihood, averaged across the $n$ tasks. In Alg. 3, we give the full procedure of selecting the lengthscale and fitting GPs to the meta-training tasks.

---

**Algorithm 3** Fitting the GPs to the meta tasks

**Input:** $n$ datasets $\{\mathcal{D}_i\}_{i=1}^n$, where $\mathcal{D}_i = \{(\mathbf{x}_{ij}, y_{ij})\}_{j=1}^{m_i}$
**Input:** set of $p$ Matérn lengthscale value candidates $\mathcal{L} = \{\ell_j\}_{j=1}^p$
**Input:** zero-mean GP prior $\mathcal{P}(f) = \mathcal{GP}\left(f \mid k_\ell\left(\mathbf{x}, \mathbf{x}'\right)\right)$, specified by the Matérn kernel $k_\ell : \mathcal{X} \times \mathcal{X} \to \mathbb{R}$
**for** $i = 1, \ldots, n$ **do**
$\quad \{\mathcal{GP}(f \mid k_{\ell_j})_i\}_{j=1}^p, \{\text{score}_{i,j}\}_{j=1}^p = CV_{4\text{-fold}}(\mathcal{D}_i, \mathcal{P})$      // cross-validation on $\mathcal{D}_i$
$j^* = \arg\max_{j=1,\ldots,p} \frac{1}{n} \sum_{i=1}^n \text{score}_{i,j}$      // selecting the optimal kernel parameters
**Output:** $n$ GP posteriors $\{\mathcal{GP}(f \mid k_{(\ell,\nu)_{j^*}})\}_{i=1}^n$

---

**Interpolation with Bayesian Neural Networks.** For MARS-BNN, in order to interpolate the datasets, we use the Monte-Carlo dropout (MC-dropout) approach by Gal & Ghahramani (2016) which trains neural networks with dropout and also uses dropout at inference/test time to generate random forward passes through the network.

Consider a neural network with $L$ layers. For each layer $l$, let us denote as $\mathbf{M}_l$ the (random) weight matrix of dimensions $K_l \times K_{l-1}$. MC-dropout samples these weight matrices by randomly dropping

rows:

$$\mathbf{M}_l = \mathbf{W}_l \cdot \mathrm{diag}\left([\mathbf{z}_{l,j}]_{j=1}^{K_i}\right)$$

$$\mathbf{z}_{l,j} \sim \mathrm{Bernoulli}\,(p_i) \ \text{for } l = 1, \ldots, L, j = 1, \ldots, K_{i-1}$$

given some dropout probabilities $p_l$ and trained weight matrices $\mathbf{W}_l$ as variational parameters. We write $\boldsymbol{\omega} = \{\mathbf{W}_l\}_{l=1}^L$ for the set variational parameters. The binary variable $\mathbf{z}_{l,j} = 0$ corresponds to neuron $j$ in layer $l$ being dropped. We can view these random weights $\mathbf{M}_l$ as samples from some variational distribution with parameters $\boldsymbol{\omega}$.

For every dataset $\mathcal{D}_i$ we fit the respective MC-dropout BNNs to the dataset $\mathcal{D}_i$, resulting in a set of variational parameters $\boldsymbol{\omega}_i$. This yields the posterior distribution $\tilde{p}(\mathbf{f}_i^{\mathbf{X}}|\mathbf{X}, \boldsymbol{\omega}_i)$ over function values $\mathbf{f}_i^{\mathbf{X}}$ for arbitrary measurement sets $\mathbf{X}$. Sampling from $\tilde{p}(\mathbf{f}_i^{\mathbf{X}}|\mathbf{X}, \boldsymbol{\omega}_i)$ corresponds to a random (i.e., with randomly dropped neurons) forward pass with inputs $\mathbf{X}$. Analogously to (3) case, the score matching loss modifies to

$$\tilde{\mathcal{L}}(\phi) := \mathbb{E}_{\mathbf{X}}\left[\frac{1}{n}\sum_{i=1}^n \mathbb{E}_{\tilde{p}(\tilde{\mathbf{f}}_i^{\mathbf{X}}|\mathbf{X}, \boldsymbol{\omega}_i)}\left[\mathrm{tr}(\nabla_{\tilde{\mathbf{f}}_i^{\mathbf{X}}}\mathbf{s}_\phi(\tilde{\mathbf{f}}_i^{\mathbf{X}}, \mathbf{X})) + \frac{1}{2}||\mathbf{s}_\phi(\tilde{\mathbf{f}}_i^{\mathbf{X}}, \mathbf{X})||_2^2\right]\right] \ . \tag{7}$$

In order to approximate the inner expectation in Eqn. (7), we sample one realisation of $\tilde{p}(\mathbf{f}_i^{\mathbf{X}}|\mathbf{X}, \boldsymbol{\omega}_i)$ per iteration by performing a random forward pass through the BNN. We fix the dropout probabilities of all layers $p_1 = \ldots = p_L = p$, across all datasets to a value $p$, which is a tunable hyperparameter.

In the experiments, we use an fully connected neural network with 3 hidden layers of size 32 each. As the activation, we use the leaky-relu. To train each network is we minimize the MSE and with the Adam optimizer with weight decay.. Weights and bias initialization are set to default initializations by Haiku (Hennigan et al., 2020), i.e., truncated normal distiribution for the weights and constant initialization for the biases.

We fix the number of epochs to 100. The hyper-parameters are selected from the following:

- learning rate: $\{1e^{-2}, 1e^{-3}, 1e^{-4}, 5e^{-3}, 5e^{-4}\}$
- weight decay: $\{0., 1e^{-2}, 1e^{-3}, 1^{e-4}, 5e^{-3}, 5e^{-4}\}$
- batch size: $\{1, 2, 4, 8, 16\}$
- dropout: $\{0.1, 0.25, 0.5, 0.7, 0.8\}$

Note that for Physionet dataset, as the datasets are varying and some have less than 16 data points, if the selected batch size is 16 and the dataset contains less than 16 points, we select the full dataset.

### A.5 Choosing the measurement distribution $\nu$

The measurement distribution $\nu$ should be supported on relevant parts of the domain $\mathcal{X}$ from which we may see queries at test time. In our experiments, we choose $\nu = \mathcal{U}(\tilde{\mathcal{X}})$ as uniform distribution over the hypercube $\tilde{\mathcal{X}} \subseteq \mathcal{X} \subseteq \mathbb{R}^d$ which conservatively covers the data. In particular for each dimension $k = 1, \ldots, d$ we compute the minimum and maximum value that occurs in the meta-training data, i.e.,

$$x_{\min}^{(k)} = \min_{i=1,\ldots,n} \min_{j=1,\ldots,m_i} x_{i,j}^{(k)}, \quad x_{\max}^{(k)} = \max_{i=1,\ldots,n} \max_{j=1,\ldots,m_i} x_{i,j}^{(k)},$$

and expand the respective ranges by 20% on each side

$$x_{\mathrm{low}}^{(k)} = x_{\min}^{(k)} - 0.2 \cdot (x_{\max}^{(k)} - x_{\min}^{(k)}), \quad x_{\mathrm{high}}^{(k)} = x_{\max}^{(k)} + 0.2 \cdot (x_{\max}^{(k)} - x_{\min}^{(k)}) \ .$$

We construct the hypercube $\tilde{\mathcal{X}} = \left[x_{\mathrm{low}}^{(1)}, x_{\mathrm{high}}^{(1)}\right] \times \ldots \times \left[x_{\mathrm{low}}^{(d)}, x_{\mathrm{high}}^{(d)}\right]$ from the Cartesian product of the expand ranges $\left[x_{\mathrm{low}}^{(k)}, x_{\mathrm{high}}^{(k)}\right]$.

## B  Meta-Learning Environments

In the following, we provide details on the meta-learning environments used in our experiments in Section 6. We list the number of tasks and samples in each environment in Table 5.

| | Sinusoid | SwissFEL | Physionet | Berkeley | Berkeley* | Argus-Control |
|---|---|---|---|---|---|---|
| $n$ | 20 | 5 | 100 | 10 | 36 | 20 |
| $m_i$ | 8 | 200 | 4 - 24 | 30 | 288 | 500 |

Table 5: Number of tasks $n$ and samples per task $m_i$ for the different meta-learning environments.

## B.1 SINUSOIDS (SYNTHETIC ENVIRONMENT)

The sinusoid environment corresponds to a simple 1-dimensional regression problem with a sinusoidal structure. It is used for visualization purposes in Figure 1 and Figure 3a. Each task of the sinusoid environment corresponds to a parametric function

$$f_{a,b,c,\beta}(x) = \beta * x + a * \sin(1.5 * (x - b)) + c , \tag{8}$$

yielding a sum of affine and a sinusoid function. Tasks differ in the function parameters $(a, b, c, \beta)$ that are sampled from the task environment $\mathcal{T}$ as follows:

$$a \sim \mathcal{U}(0.7, 1.3), \quad b \sim \mathcal{N}(0, 0.1^2), \quad c \sim \mathcal{N}(5.0, 0.1^2), \quad \beta \sim \mathcal{N}(0.5, 0.2^2) . \tag{9}$$

Figure 3a displays functions $f_{a,b,c,\beta}$ with parameters sampled according to (9). To draw training samples from each task, we uniformly sample $x$ from $\mathcal{U}(-5, 5)$ and add Gaussian noise with standard deviation 0.1 to the function values $f(x)$:

$$x \sim \mathcal{U}(-5, 5) , \qquad y \sim \mathcal{N}(f_{a,b,c,\beta}(x), 0.1^2) . \tag{10}$$

## B.2 SWISSFEL

Free-electron lasers (FELs) accelerate electrons to a very high speed to generate shortly pulsed laser beams with wavelengths in the X-ray spectrum. The X-ray pulses from the accelerator can map nanometer-scale structures, thus facilitating molecular biology and material science experiments. The accelerator and the electron beam line of an FEL consist of multiple magnets and other adjustable components, which have several parameters that experts adjust in order to maximize the pulse energy (Kirschner et al., 2019a). Due to different operational modes, parameter drift, and changing (latent) conditions, the laser's pulse energy function, in response to its parameters, changes across time. As a result, optimizing the laser's parameters is a recurrent task.

**Meta-learning setup.** The meta-learning environment represents different parameter optimization runs (i.e., tasks) on SwissFEL, an 800-meter-long free electron laser located in Switzerland (Milne et al., 2017). The input space is 12-dimensional and corresponds to the laser parameters, whereas the regression target corresponds to the one-dimensional pulse energy. We refer to Kirschner et al. (2019b) for details on the individual parameters. Each optimization run consists of roughly 2000 data points generated with online optimization methods, yielding non-i.i.d. data, which becomes successively less diverse throughout the optimization. To avoid issues with highly dependent data points, we take the first 400 data points per run and split them into training and test subsets of size 200. As we have a total of 9 runs (tasks) available, we use 5 of them for meta-training and the remaining 4 for meta-testing.

## B.3 PHYSIONET

In the context of the Physionet competition 2012, Silva et al. (2012) have published an open-access dataset of patient stays in the intensive care unit (ICU). The dataset consists of measurements taken during the patient stays, where up to 37 clinical variables are measured over the span of 48 hours, yielding a time series of measurements. The intended task for the competition was the binary classification of patient mortality. However, the dataset is also often used for time series prediction methods due to a large number of missing values (around 80 % across all features).

**Meta-learning setup.** To set up the meta-learning environment, we treat each patient as a separate task and the different clinical variables as different environments. Out of the 37 variables, we picked the Glasgow coma scale (GCS) and hematocrit value (HCT) as environments for our study since

|  | RMSE | | Calib. error | |
|---|---|---|---|---|
|  | Full dataset | Partial dataset | Full dataset | Partial dataset |
| Vanilla GP | $0.276 \pm 0.000$ | $0.258 \pm 0.000$ | $\mathbf{0.109 \pm 0.000}$ | $0.119 \pm 0.000$ |
| Vanilla BNN | $0.109 \pm 0.004$ | $0.151 \pm 0.018$ | $0.179 \pm 0.002$ | $0.206 \pm 0.025$ |
| MAML | $\mathbf{0.045 \pm 0.003}$ | $\mathbf{0.121 \pm 0.027}$ | / | / |
| BMAML | $0.073 \pm 0.014$ | $0.222 \pm 0.032$ | $0.161 \pm 0.013$ | $0.154 \pm 0.021$ |
| NP | $0.079 \pm 0.014$ | $0.173 \pm 0.018$ | $0.210 \pm 0.000$ | $0.140 \pm 0.035$ |
| MLAP | $0.050 \pm 0.034$ | $0.348 \pm 0.034$ | $\mathbf{0.108 \pm 0.024}$ | $0.183 \pm 0.017$ |
| PACOH-NN | $0.130 \pm 0.009$ | $0.160 \pm 0.070$ | $0.167 \pm 0.005$ | $0.223 \pm 0.012$ |
| MARS | $0.093 \pm 0.002$ | $\mathbf{0.116 \pm 0.024}$ | $0.140 \pm 0.002$ | $\mathbf{0.080 \pm 0.005}$ |

Table 6: Prediction accuracy and uncertainty calibration on full and partial Berkeley-Sensor dataset. On the full dataset, MARS performance is less competitive due to the strong auto-correlation of the data which is not taken into account in the BNN likelihood. On the partial dataset, which has less dependency among the data points, MARS outperforms all other meta-learning methods.

they are among the most frequently measured variables in this dataset. From the dataset, we remove all patients where less than four measurements of CGS (and HCT, respectively) are available. From the remaining patients, we used 100 patients for meta-training and 500 patients for meta-validation and meta-testing. Since the number of available measurements differs across patients, the number of training points $m_i$ ranges between 4 and 24.

### B.4 BERKELEY-SENSOR

The Berkeley dataset contains data from 46 temperature sensors deployed in different locations at the Intel Research lab in Berkeley (Madden, 2004). The temperature measurements are taken over four days and sampled at 10-minute intervals. Each task corresponds to one of the 46 sensors and requires auto-regressive prediction, particularly predicting the subsequent temperature measurement given the last ten measurements.

**Meta-learning setup.** The Berkeley environment, as used in Rothfuss et al. (2021a), uses 36 sensors (tasks) with data for the first two days for meta-training and the last 10 for meta-testing. The meta-training and meta-testing are separated temporally and spatially since the data is non-i.i.d. Data are abundant, and the measurements are taken at very close intervals. Thus, the features and the train/context data points are strongly correlated, violating the i.i.d. assumption that underlies our factorized Gaussian likelihood and Bayes rule in Section 3, causing the BNN to over-weight the empirical evidence and making over-confident predictions. To alleviate this problem, we subsample the data. In particular, we randomly select 10 out of the 36 training tasks, and instead of using all measurements, we randomly sample 30 of them. This has two effects: First, it makes the data less dependent/correlated and thus more compatible with our Bayesian formulation. Second, we increase the epistemic uncertainty by using less data, making the calibration metrics more meaningful. The results reported in Section 6 correspond to the sub-sampled data.

For completeness, we also report the results for the full dataset as in Rothfuss et al. (2021a) in Table 6. Other meta-learning baselines, such as MAML or MLAP, perform better than MARS on the full dataset since they do not explicitly use the Bayes rule with i.i.d. assumption or weight the empirical evidence less. Note that MARS performs worse due to the Bayesian inference at meta-test time rather than our meta-learning approach. Accounting for the strong auto-correlation of the data in the likelihood would most likely resolve the issue. On the sub-sampled environment, MARS again performs best.

### B.5 ARGUS-CONTROL

The final environment we use in our experiments is a robot case study. In particular, it aims at tuning the controller of the Argus linear motion system by Schneeberger Linear Technology. The goal is to choose the controller parameters so that the position error is minimal. In our setup, each task is a regression problem where the goal is to predict the total variation (TV) of the robot's position error

signal when controlled by a PID controller in simulation. The regression features are the three PID controller gain parameter parameters.

**Meta-learning setup.** Overall, the environment consists of 24 tasks, of which 20 are used for meta-training and the remaining 4 for meta-testing. Each task corresponds to a different step size for the robot to move, ranging from $10\mu m$ to $10mm$. At different scales, the robot behaves differently in response to the controller parameters, resulting in different target functions. This presents a good environment for transferring similarities across different scales while leaving enough flexibility in the prior to adjust to the target function at a step size.

## C    EXPERIMENTAL METHODOLOGY

In the following, we describe our experimental methodology used in Section 6.

### C.1    OVERVIEW OF THE META-TRAINING AND META-TESTING PHASES

Evaluating the performance of a meta-learner consists of two phases, *meta-training* and *meta-testing*. The latter phase, *meta-testing*, can be further split into *target training* and *target testing*. In particular, for MARS the phases consist of the following:

- *Meta-training:* The meta-training datasets $\mathcal{D}_{i=1}^{n}$ are used to train the score estimator network (see Algorithm 1).

- *Target training:* Equipped with knowledge about the underlying data-generation process, i.e., the score network, we perform BNN inference on a new target task with a corresponding context dataset $\mathcal{D}^*$. In particular, we run fSVGD with the score network as a swap-in for the marginal scores of the stochastic process prior (see Algorithm 2). As a result, we obtain a set of NN particles that approximates the BNN posterior in the function space.

- *Target testing:* Finally, we evaluate the approximate posterior predictions on a test set $\mathcal{D}^\dagger$ corresponding to the same target task. In particular, we compute the residual mean squared error (RMSE) and the calibration error as performance metrics. We describe the evaluation metrics in more detail in Section C.2.

The target training and testing are performed independently with the meta-learned score network for each test task. The metrics are reported as averages over the test tasks.

The entire meta-training and meta-testing procedure are repeated for five random seeds that influence the score network initialization, the sampling-based estimates in Algorithm 1 as well as the initialization of the BNN particles for target training. The reported averages and standard deviations are based on the results obtained for different seeds.

### C.2    EVALUATION METRICS

During *target-testing*, we evaluate the posterior predictions on a held-out test dataset $\mathcal{D}^\dagger$. Among the methods employed in Section 6, MARS, PACOH-NN, NPs, MLAP, Vanilla BNN and Vanilla GP yield probabilistic predictions $\hat{p}(y^\dagger | x^\dagger, \mathcal{D}^*)$ for the test points $x^\dagger \in \mathcal{D}^\dagger$. For instance, in the case of MARS, PACOH-NN, and Vanilla BNN where the posterior is approximated by a set of NN particles $\{\theta_1, ..., \theta_L\}$ and we use a Gaussian likelihood, the predictive distribution is an equally weighted mixture of Gaussians:

$$\hat{p}(y^\dagger | x^\dagger, \mathcal{D}^*) = \frac{1}{L} \sum_{l=1}^{L} \mathcal{N}(y^\dagger | h_{\theta_l}(x^\dagger), \sigma^2) \ . \tag{11}$$

The respective mean prediction corresponds to the expectation of $\hat{p}$, that is $\hat{y} = \hat{\mathbb{E}}(y^* | x^*, \mathcal{D}^*)$. In the case of MAML, only the mean prediction is available.

|  | RMSE | | | |
| --- | --- | --- | --- | --- |
|  | GP-2D | GP-3D | TP-2D | TP-3D |
| KEF | $8.836 \pm 0.000$ | $6.625 \pm 0.000$ | $8.852 \pm 0.000$ | $13.694 \pm 0.000$ |
| NKEF | $4.322 \pm 0.000$ | $4.402 \pm 0.000$ | $5.936 \pm 0.000$ | $8.402 \pm 0.000$ |
| KEF-CG IMQ* | $3.009 \pm 0.000$ | $5.197 \pm 0.000$ | $6.093 \pm 0.000$ | $5.473 \pm 0.000$ |
| KEF-CG RBF* | $1.145 \pm 0.000$ | $2.237 \pm 0.000$ | $2.869 \pm 0.000$ | $4.370 \pm 0.000$ |
| $\nu$-method IMQ* | $4.682 \pm 0.000$ | $9.171 \pm 0.000$ | $8.019 \pm 0.000$ | $10.81 \pm 0.000$ |
| $\nu$-method IMQp* | $4.682 \pm 0.000$ | $9.171 \pm 0.000$ | $11.53 \pm 0.000$ | $5.872 \pm 0.000$ |
| $\nu$-method RBF* | $1.718 \pm 0.000$ | $4.035 \pm 0.000$ | $4.278 \pm 0.000$ | $5.200 \pm 0.000$ |
| SSGE | $1.166 \pm 0.000$ | $1.989 \pm 0.000$ | $1.753 \pm 0.000$ | $4.822 \pm 0.000$ |
| Stein estimator | $0.912 \pm 0.000$ | $4.058 \pm 0.000$ | $3.519 \pm 0.000$ | $7.251 \pm 0.000$ |
| MARS network | $\mathbf{0.664 \pm 0.179}$ | $\mathbf{1.375 \pm 0.215}$ | $\mathbf{1.093 \pm 0.241}$ | $\mathbf{3.199 \pm 0.418}$ |

Table 7: RMSE between the true and predicted scores of two/three-dimensional marginal distributions of GP and TP with an RBF kernel and a sinusoidal mean function with a linear trend. MARS score network significantly outperforms all nonparametric score estimators. Estimators with a $^*$ correspond to curl-free estimators (Zhou et al., 2020) with either an IMQ or RBF kernel.

**Evaluating prediction accuracy (RMSE)**  Based on the mean predictions, we compute the *root-mean-squared error (RMSE)*

$$\text{RMSE} = \sqrt{\frac{1}{|\mathcal{D}^\dagger|} \sum_{(x^\dagger, y^\dagger) \in \mathcal{D}^\dagger} (y^\dagger - \hat{\mathbb{E}}(y^\dagger | x^\dagger, \mathcal{D}^*))^2} \qquad (12)$$

which quantifies how accurate the mean predictions are.

**Evaluating uncertainty calibration (Calibration error)**  In addition to the prediction accuracy, we also assess the quality of the uncertainty estimates. For this purpose, we use the concept of calibration, i.e., a probabilistic predictor is calibrated if the predicted probabilities are consistent with the observed frequencies on unseen test data. We use a regression calibration error similar to Kuleshov et al. (2018) in order to quantify how much the predicted probabilities deviate from the empirical frequencies.

Let us denote the predictor's cumulative density function (CDF) as $\hat{F}(y \mid \mathbf{x}, \mathcal{D}^*) = \int_{-\infty}^y \hat{p}(\tilde{y}|\mathbf{x}, \mathcal{D}^*)d\tilde{y}$, where $\hat{p}(\tilde{y}|\mathbf{x}, \mathcal{D}^*)d\tilde{y}$ is the predictive posterior distribution. For confidence levels $0 \le q_h < \ldots < q_H \le 1$, we compute the corresponding empirical frequency

$$\hat{q}_h = \frac{\left| \left\{ y^\dagger \mid \hat{F}\left(y^\dagger \mid \mathbf{x}, \mathcal{D}^*\right) < q_h, \, (\mathbf{x}^\dagger, y^\dagger) \in \mathcal{D}^\dagger \right\} \right|}{|\mathcal{D}^\dagger|},$$

based on some test dataset $\mathcal{D}^\dagger$. If the predictions are calibrated, we expect that $\hat{q}_h \to q_h$ as $m \to \infty$. Following Kuleshov et al. (2018), we define the calibration error metric as a function of the residuals $\hat{q}_h - q_h$:

$$\text{calib-err} = \frac{1}{H} \sum_{h=1}^{H} |\hat{q}_h - q_h|.$$

Note that we report the average of absolute residuals $|\hat{q}_h - q_h|$, rather than reporting the average of squared residuals $|\hat{q}_h - q_h|^2$, as done by Kuleshov et al. (2018). This is done to preserve the units and keep the calibration error easier to interpret. In our experiments, we compute the empirical frequency using $M = 20$ equally spaced confidence levels between 0 and 1.

### C.3  HYPER-PARAMETER SELECTION

For each of the meta-environments and algorithms, we ran a separate hyper-parameter search to select the hyper-parameters. In particular, we use 30 randomly sampled hyperparameter configurations across five randomly selected seeds and select the best-performing one in terms of either RMSE or

| | Cosine similarity | | | |
|---|---|---|---|---|
| | GP-2D | GP-3D | TP-2D | TP-3D |
| KEF | $0.475 \pm 0.000$ | $0.410 \pm 0.000$ | $0.532 \pm 0.000$ | $0.485 \pm 0.000$ |
| NKEF | $0.556 \pm 0.000$ | $0.434 \pm 0.000$ | $0.384 \pm 0.000$ | $0.318 \pm 0.000$ |
| KEF-CG IMQ* | $0.286 \pm 0.000$ | $0.308 \pm 0.000$ | $0.303 \pm 0.000$ | $0.538 \pm 0.000$ |
| KEF-CG RBF* | $0.621 \pm 0.000$ | $0.457 \pm 0.000$ | $0.479 \pm 0.000$ | $0.428 \pm 0.000$ |
| $\nu$-method IMQ* | $0.337 \pm 0.000$ | $0.286 \pm 0.000$ | $0.251 \pm 0.000$ | $0.384 \pm 0.000$ |
| $\nu$-method IMQp* | $0.337 \pm 0.000$ | $0.286 \pm 0.000$ | $0.374 \pm 0.000$ | $0.411 \pm 0.000$ |
| $\nu$-method RBF* | $0.809 \pm 0.000$ | $0.759 \pm 0.000$ | $0.628 \pm 0.000$ | $0.786 \pm 0.000$ |
| SSGE | $\mathbf{0.943 \pm 0.000}$ | $0.601 \pm 0.000$ | $\mathbf{0.661 \pm 0.000}$ | $0.521 \pm 0.000$ |
| Stein estimator | $0.778 \pm 0.000$ | $\mathbf{0.851 \pm 0.000}$ | $0.626 \pm 0.000$ | $0.590 \pm 0.000$ |
| MARS network | $\mathbf{0.956 \pm 0.110}$ | $\mathbf{0.889 \pm 0.048}$ | $\mathbf{0.737 \pm 0.097}$ | $\mathbf{0.706 \pm 0.061}$ |

Table 8: Cosine Similarity between the true and predicted scores of two/three-dimensional marginal distributions of GP and TP with an RBF kernel and a sinusoidal mean function with a linear trend. MARS score network performs competitively to nonparametric score estimators. Estimators with a * correspond to curl-free estimators (Zhou et al., 2020) with either an IMQ or RBF kernel.

calibration error. For the reported results, we provide the chosen hyperparameters and detailed evaluation results in `https://tinyurl.com/376wp8xe`. The code scripts for reproducing the experimental results are provided in our repository[2].

## D  FURTHER EXPERIMENT RESULTS

We provide further empirical evidence for the proposed method. In particular, we showcase that:

1. Our parametric score matching approach performs favorably to many nonparametric score estimators.
2. Regularization via spectral normalization does not hinder the flexibility of the network.
3. Sampling from the GP posteriors when training the score network successfully incorporates epistemic uncertainty in areas of the domain where meta-training data is scarce.

### D.1  PARAMETRIC VS NONPARAMETRIC SCORE ESTIMATION METHODS

We start by comparing the performance of MARS to that of several nonparametric score estimators, described in Zhou et al. (2020).

**Nonparametric score estimators.**  There are several theoretically motivated nonparametric score estimation methods with well-understood properties and straightforward implementations. Their simplicity and flexibility make them a popular choice (Coyle et al., 2020; Wang et al., 2019; Deledalle et al., 2014). KEF (Canu & Smola, 2006) performs regularized score matching inside a kernel exponential family; this can further be efficiently approximated by the Nyström method (NKEF) (Sutherland et al., 2018). Liu et al. (2016); Chwialkowski et al. (2016) propose a Stein estimator, and Shi et al. (2018) propose Spectral Stein Gradient Estimator (SSGE) by expanding the score function in terms of the spectral eigenbasis. Together with iterative methods (the $\nu$-method and Landweber iteration (Engl et al., 1996)), these approaches can be naturally unified through a regularized, nonparametric regression framework (Zhou et al., 2020), in which the conjugate-gradient version of the KEF estimator is also proposed (KEF-CG). In our evaluations, we reports results for curl-free sccore estimators with both an inverse multiquatric (IMQ) or RBF kernel.

**MARS vs nonparametric estimators.**  To compare the performance of MARS to nonparametric methods, we consider score estimation of marginal distributions of Gaussian Processes (GP) (Rasmussen, 2003) and Student's-$t$ processes (TP) (Shah et al., 2014). In both cases, as the stochastic

---

[2]https://github.com/krunolp/mars

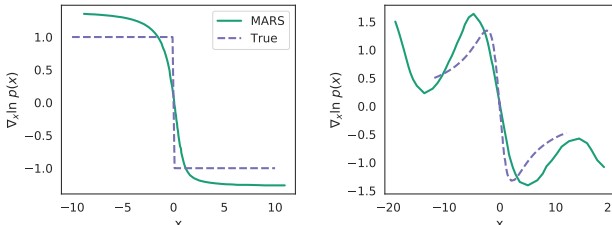

Figure 5: Predictions of the MARS score network, trained on 45 samples from Laplace (left) and Student's-$t$ distribution (right). The network approximates the score functions sufficiently well, showcasing that spectral normalization of linear layers does not hinder its flexibility.

process mean function, we use the sinusoidal mean function $2x + 5sin(2x)$, and for the covariance kernel, we choose the radial basis function kernel, with the lengthscale parameter set to $\ell = 1$. We use Tensorflow Probability's (Dillon et al., 2017) implementation of both GP and TP and set all other parameters to their default values.

We perform four experiments, the first two on estimating the marginal scores of a GP and the last two on the TP. In all experiments, we sample one measurement set $\mathbf{X}$ containing either two ($\mathbf{X} = \{x_1, x_2\}$) or three points ($\mathbf{X} = \{x_1, x_2, x_3\}$) of dimension one, i.e., $x_i \in \mathbb{R}, \forall i$. For all experiments, the $x_i$ follow a uniform distribution: for the GP experiment, $x_i \sim U([-5,5])$ , whereas for the TP experiment, $x_i \sim U([-1,1]), \forall i$. In the experiments with measurement sets consisting of two points, the score network and the nonparametric estimators are trained on 50 samples from the corresponding *two*-dimensional marginal distributions. For the measurement sets of size 3, we train the score network and nonparametric estimators on 200 samples from the corresponding *three*-dimensional marginal distribution. When evaluating the performance of MARS, we take the average performance over five different seeds after training the network for 2000 iterations in the *two*-dimensional marginal case and 5000 iterations in the *three*-dimensional case. We measure the quality of the score estimates via the RMSE and cosine similarity between the estimates and true scores. The corresponding evaluation results are reported in Tables 7-8.

The RMSE measures deviations of the estimated scores in both direction and magnitude of the gradients. For the SVGD particle estimation, it is more important that the gradients in the vector field point in the correct direction than having the correct magnitude. Thus, we also report the cosine similarity, which only quantifies how well the directions in the estimated vector field match the directions in the true vector field while neglecting errors in the score magnitude. As we can see in Table 7 and 8, MARS consistently outperforms the nonparametric score estimators. Among the nonparametric methods, SSGE performs best on average. Hence, we use SSGE in our ablation study in Section 6.3.

## D.2 FLEXIBILITY OF THE REGULARIZED NETWORK

Recall from section 5.3 that, in order to prevent overfitting, we perform spectral normalization of the weights of the linear layers by re-parameterizing the weights by $\tilde{\mathbf{W}} := \mathbf{W}/||\mathbf{W}||$. To investigate whether this hinders the flexibility of the network's outputs, we visually examine the network's predictions on the following two tasks, in which we estimate the scores of one-dimensional Laplace and Student's-$t$ distribution.

In order to estimate a score $\nabla_x \log p(x)$ for some unknown distribution $p(x)$ (rather than marginal distribution scores $\nabla_{\mathbf{f}^{\mathbf{X}}} \ln p(\mathbf{f}^{\mathbf{X}})$ for some unknown stochastic process $p(f)$), we use 45 samples $\mathbf{X} = \{x_1, ..., x_{45}\}$ from: *(i)* one-dimensional Student's-$t$ distribution with $\nu = 5$ degrees of freedom, location parameter $\mu = 0$, and scale parameter $\sigma = 1$, and *(ii)* one-dimensional standard Laplace distribution, i.e., with location parameter $\mu = 0$ and scale parameter $\sigma = 1$.

To perform distributional score estimation, we need to make a slight modification to the original score network architecture; rather than concatenating input-output pairs (i.e., concatenating $x_i$ and $f(x_i)$, as shown in Fig. 4), we use only the distributional samples as inputs. To be more precise, at every iteration, we randomly select (without replacement) $k = 8$ inputs $\{x'_1, ..., x'_k\} \subset \mathbf{X}$, which we input to the score network. We train the network using the regularized score matching loss, i.e.,

perform spectral normalization of the layers. The network is trained for 2000 iterations, using a learning rate of 0.001. For this experiment, we set the attention embedding dimension to 32, key size to 16, and use 8 attention heads. The remaining hyperparameters are set according to Sec. A.3. The respective predictions are displayed in Figure 5. We observe that the score network approximates the score functions well and that the regularization does not hinder flexibility too much.

## D.3 INCORPORATING UNCERTAINTY THROUGH GP INTERPOLATION

In addition to the ablation study performed in Section 6.3, we visually investigate the implications of sampling functions from the GP posteriors during score matching. In particular, we do so using the sinusoid environment, detailed in Appendix B.1.

**Experiment setup.** In this experiment, we sample 10 functions from the environment and evaluate each function at five randomly selected inputs in the $[-5, -2] \cup [2, 5]$ range. We investigate whether the proposed GP interpolation method promotes uncertainty in the $[-2, 2]$ part of the domain.

In order to do so, let us recall Algorithm 1, in which we learn the trained score network $\mathbf{s}_\phi$. The algorithm first fits a GP to each of the $n$ datasets $\mathcal{D}_1, ..., \mathcal{D}_n$. Then, at every step, the algorithm samples a measurement set $\mathbf{X} \overset{iid}{\sim} \nu$, and then, for every dataset $\mathcal{D}_i$ (i.e., for every collection of input/output pairs $\mathbf{X}_i^{\mathcal{D}}, \mathbf{y}_i^{\mathcal{D}}$), the algorithm samples function values from the GP posterior marginal: $\tilde{\mathbf{f}}_i^{\mathbf{X}} \sim p(\mathbf{f}_i^{\mathbf{X}} | \mathbf{X}, \mathbf{X}_i^{\mathcal{D}}, \mathbf{y}_i^{\mathcal{D}})$. We compare this approach to training the score network $\mathbf{s}_\phi$ on the mean predictions of the posterior marginal $p(\mathbf{f}_i^{\mathbf{X}} | \mathbf{X}, \mathbf{X}_i^{\mathcal{D}}, \mathbf{y}_i^{\mathcal{D}})$, i.e., where $\tilde{\mathbf{f}}_i^{\mathbf{X}} = \mu_{p(\mathbf{f}_i^{\mathbf{X}} | \mathbf{X}, \mathbf{X}_i^{\mathcal{D}}, \mathbf{y}_i^{\mathcal{D}})}$. In order to distinguish between the two approaches, we denote the score network trained on the posterior marginal mean predictions with $\mathbf{s}_\phi^\mu$, and use $\tilde{\mathbf{f}}_{\mu\ i}^{\mathbf{X}} := \mu_{p(\mathbf{f}_i^{\mathbf{X}} | \mathbf{X}, \mathbf{X}_i^{\mathcal{D}}, \mathbf{y}_i^{\mathcal{D}})}$ as a shorthand notation for the mean predictions of the posterior marginals. Observe that, for measurement sets $\mathbf{X}$ close to $\mathbf{X}_i^{\mathcal{D}}$, the samples $\tilde{\mathbf{f}}_i^{\mathbf{X}}$ and $\tilde{\mathbf{f}}_{\mu\ i}^{\mathbf{X}}$ will be very similar, whereas when $\mathbf{X}$ is far from $\mathbf{X}_i^{\mathcal{D}}$, the variability of the samples $\tilde{\mathbf{f}}_i^{\mathbf{X}}$ will be much larger. In the following, we showcase that this variability successfully incorporates uncertainty about the areas of the domain $\mathcal{X}$ with little or no data.

In order to compare the two approaches, we visually compare the overall framework when the fSVGD network is trained with $\mathbf{s}_\phi^\mu$ (the score network trained on $\tilde{\mathbf{f}}_{\mu\ i}^{\mathbf{X}}$, the GP mean predictions), and when it is trained with $\mathbf{s}_\phi$ (the score network trained on $\tilde{\mathbf{f}}_i^{\mathbf{X}}$, the samples from the GP posterior marginal), as performed in the original MARS algorithm.

**Empirical findings.** We plot the corresponding fSVGD-BNN (fBNN) predictions, trained using the two score networks $\mathbf{s}_\phi^\mu$ and $\mathbf{s}_\phi$. Both fBNNs are fitted to four points, where inputs $x_1, ..., x_4$ are sampled uniformly from $[-5, 5]$ (i.e., the whole input domain), and their functional evaluations are obtained according to the sinusoid environment.

The results are visualized in Figure 6. The first two plots on the left in Figure 6 correspond to posterior predictions of two randomly selected GPs fitted to the meta-tasks, where no task contains information in the $[-2, 2]$ input range. The middle plot corresponds to the fBNN network predictions, where the network was trained using the fSVGD algorithm and the score network $\mathbf{s}_\phi^\mu$. The last plot corresponds to the fBNN network predictions, where fBNN is trained using the fSVGD algorithm and the score network $\mathbf{s}_\phi$. We observe a clear difference between the two approaches: MARS (trained using $\mathbf{s}_\phi$) successfully incorporates the epistemic uncertainty in the $[-2, 2]$ part of the input domain into the fBNN posterior, yielding less confident predictions in the area where no data was available during meta-training. In contrast, when we use GP posterior means instead of samples when fitting the score network, the resulting BNN predictions entirely ignore the epistemic uncertainty that arises due to the fact that we don't know the function values in $[-2, 2]$. This may lead to overconfident posterior predictions.

## D.4 COMPARISON OF INTERPOLATION METHODS

So far, we have experimented with using GPs (MARS-GP) and MC-dropout BNNs as interpolators (MARS-BNN) and sample random random function values from the corresponding posteriors when training our score network. In the ablation study (performed in Sec. 6.3), we also experimented with

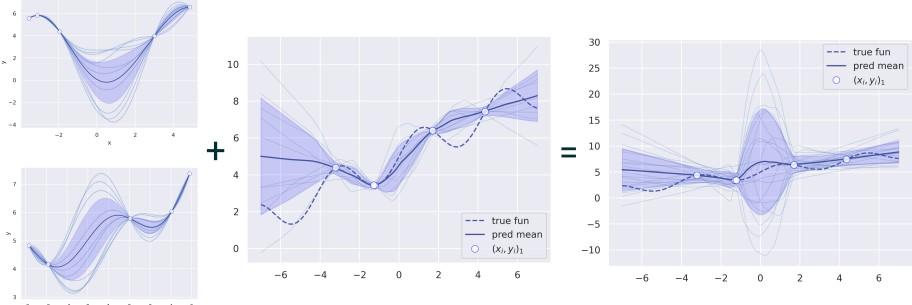

Figure 6: MARS prediction on samples from the sinusoid environment, with no data in the $[-2, 2]$ range. Left: posterior predictions of two randomly selected GPs fitted to the meta-tasks. Middle: fBNN predictions, fitted using $\mathbf{s}_\phi^\mu$, the score estimation network trained on GP mean predictions. Right: fBNN predictions, fitted using $\mathbf{s}_\phi$, the score estimation network trained on samples from the GPs. Training fBNN with $\mathbf{s}_\phi$ (by sampling from the GPs) successfully incorporates uncertainty about the $[-2, 2]$ part of the input domain.

| | SwissFEL | Physionet-GCS | Physionet-HCT | Berkeley-Sensor | Argus-Control |
|---|---|---|---|---|---|
| GP mean | $0.471 \pm 0.059$ | $2.994 \pm 0.363$ | $5.995 \pm 1.108$ | $1.253 \pm 0.112$ | $0.073 \pm 0.003$ |
| MARS-GP | $\mathbf{0.391 \pm 0.011}$ | $1.471 \pm 0.083$ | $\mathbf{2.309 \pm 0.041}$ | $\mathbf{0.116 \pm 0.024}$ | $\mathbf{0.013 \pm 0.001}$ |
| determ. NN | $\mathbf{0.380 \pm 0.032}$ | $2.891 \pm 0.042$ | $2.530 \pm 0.049$ | $0.306 \pm 0.068$ | $\mathbf{0.017 \pm 0.003}$ |
| NN Ensemble | $0.423 \pm 0.031$ | $1.539 \pm 0.074$ | $\mathbf{2.281 \pm 0.014}$ | $0.134 \pm 0.045$ | $\mathbf{0.015 \pm 0.001}$ |
| MARS-BNN | $\mathbf{0.407 \pm 0.061}$ | $\mathbf{1.307 \pm 0.065}$ | $\mathbf{2.248 \pm 0.057}$ | $\mathbf{0.113 \pm 0.015}$ | $\mathbf{0.017 \pm 0.003}$ |

Table 9: Comparison of various interpolation methods in terms of the RMSE. Reported are the mean and standard deviation across five seeds. MARS-BNN and MARS-GP perform the best.

using GP's mean predictions rather than sampling from the GP posterior. Here, we provide empirical analysis of using two further interpolation methods: *(i)* deterministic NNs, and *(ii)*, ensembles of NNs in our generic MARS pproach.

Both the deterministic NN and the NN ensemble interpolators are trained similarly to the MC-dropout BNNs, as described in Section A.4. For the NN ensembles, we choose the number of of ensemble members as a hyper-parameter from $\{5, 10, 20\}$. In case of the deterministic NN, we do not perform any posterior sampling in (3) and just use the NN's predictions as function values for the score estimation. In case of the ensemble, we randomly sample the function values corresponding to one ore multiple ensemble member. The number of function value samples from the ensemble is chosen as hyper-parameter and selected from $\{1, 2, 3, 4\}$. The results for employing MARS with different interpolators are reported in Table 9-10. Not surprisingly, the not accounting for epistemic uncertainty (i.e., GP mean and deterministic NN) performs much worse in the majority of environments, in particular, w.r.t. the calibration error. Among the neural network based interpolators, the MC-dropout BNNs consistently perform the best. This is why suggest their usage and evaluate them in the main part of the paper.

| | SwissFEL | Physionet-GCS | Physionet-HCT | Berkeley-Sensor | Argus-Control |
|---|---|---|---|---|---|
| GP mean | $0.204 \pm 0.013$ | $\mathbf{0.225 \pm 0.021}$ | $0.237 \pm 0.018$ | $0.141 \pm 0.029$ | $0.216 \pm 0.066$ |
| MARS-GP | $\mathbf{0.035 \pm 0.002}$ | $0.263 \pm 0.001$ | $\mathbf{0.136 \pm 0.007}$ | $\mathbf{0.080 \pm 0.005}$ | $\mathbf{0.055 \pm 0.002}$ |
| determ. NN | $0.070 \pm 0.025$ | $\mathbf{0.249 \pm 0.014}$ | $0.233 \pm 0.016$ | $0.112 \pm 0.022$ | $0.088 \pm 0.028$ |
| NN Ensemble | $0.081 \pm 0.010$ | $\mathbf{0.238 \pm 0.018}$ | $0.248 \pm 0.021$ | $0.129 \pm 0.028$ | $0.118 \pm 0.014$ |
| MARS-BNN | $0.054 \pm 0.009$ | $\mathbf{0.268 \pm 0.023}$ | $0.231 \pm 0.029$ | $\mathbf{0.078 \pm 0.020}$ | $\mathbf{0.076 \pm 0.031}$ |

Table 10: Comparison of various interpolation methods in terms of the calibration error. Reported are the mean and standard deviation across five seeds. MARS-GP performs best, with MARS-BNN second best performing.

