# OpenReview forum: "MARS: Meta-learning as Score Matching in the Function Space"
_ICLR.cc/2023/Conference — ICLR 2023 notable top 25%_

### Official Review · Reviewer_HXi8 · 2022-10-23

**Confidence:** 4
**Correctness:** 3
**Technical Novelty And Significance:** 3
**Empirical Novelty And Significance:** 3
**Recommendation:** 8

**Clarity, Quality, Novelty And Reproducibility:**

The paper is well written and easy to follow. The overall idea is novel and interesting.

**Strength And Weaknesses:**

Strength

1. The paper is well scoped and design flow is clear. Each new idea is given a clear motivation or observation.
2. The idea of fitting score function by a network for meta learning is novel and interesting.
3. The experimental results show the consistently good performance comparing benchmark methods.

Weakness

1. The authors use fSVGD rather than functional BNN, but no comparative evaluations are given in the experiments to verify this choose. Since fSVGD is particles-based, it may affect the computational efficiency comparing normal fBNN. It is expected to what the additional advantages comparing with fBNN with GP prior.

2. A transformer encoder is used to model score function considering the permutation equivariance property of the process. However, there is another property of the process that is marginal consistency. Is there any control of the encoder to ensure such property? If not, what is the possible effect when loss such property?

3. In 5.2, the authors propose to fit each task by a GP in order to use eq. (2). If you can use a GP to model each task, do you still need the score network? Can we just fit a prior for all GPs? like a Hierarchical GP? Have you tested on such option?










**Summary Of The Paper:**

This paper proposes a new Bayesian meta learning algorithm which aims to relax the traditional restricted task prior. The idea is to use a stochastic process prior for all tasks rather than plain diagonal Gaussian prior for over-parameterized Bayesian neural networks. The new prior is claimed to be with less possibility of over-fitting and improved uncertainty estimation. Further, instead of estimating the prior, the authors propose to only fit the score function by a transformer encoder and then use fSVGD in meta training and test. Specifically, a GP is fitted for each task in order to generate the values on all possible measurement set and spectral normalization is used to further reduce the tendency of overfitting.

**Summary Of The Review:**

The paper is well written and easy to follow. The overall idea is novel and interesting. There are still some weaknesses regarding the design (please see Weakness for more details).

---

> ### Author Response · Authors · 2022-11-15
> **Authors' response**
>
> Thank you for taking the time to clearly understand our paper and for the encouraging feedback. In the following, we try to address your remaining concerns.
>
> > The authors use fSVGD rather than functional BNN, but no comparative evaluations are given in the experiments to verify this choose. Since fSVGD is particles-based, it may affect the computational efficiency comparing normal fBNN.
>
> We use fSVGD as a functional approximate inference method for the following two reasons:
>
> First, we found fSVGD to work better than fBNNs at the beginning of our experiments.
>
> Second, fSVGD methods only require the estimation of the prior scores. Since we swap in our score network for the prior score, we do not need any score estimation at meta-test time which speeds up the BNN approximate inference considerably. In contrast, for fBNNs, we have to estimate both the prior score as well as the score of the approximate posterior. So, even though we have amortized estimating the prior score, we still need to use SSGE in every iteration when optimizing the functional variation inference objective. Thus, despite being particle-based, MARS + fSVGD inference is considerably faster than using MARS + fBNNs.
>
> Finally, our approach is about meta-learning stochastic process priors. Thus, it is orthogonal to the choice of approximate inference method which is used at meta-test time. Hence, our meta-learning method can be freely combined with any functional approximate inference methods such as fVI/fBNNs.
>
> > [...] there is another property of the process that is marginal consistency. Is there any control of the encoder to ensure such property? If not, what is the possible effect when loss such property?
>
> This is an excellent question to which we do not have a fully satisfying answer. Our score network architecture itself does not enforce marginal consistency. In general, marginal consistency comes into play when we look at measurement sets of different sizes. By training the score network with varying measurement set sizes, we automatically push the score networks toward marginal consistency. The score matching loss is consistent (e.g. see [1]). Thus, in the asymptotic data limit when we have a large number of tasks and samples per task, and, when the score network is sufficiently large/expressive, the score predictions converge to the true marginal scores and we attain marginal consistency.
>
> In practice, with finite width/depth, regularization, and finite data, our score network will most likely not be marginally consistent. However, this is generally not a big issue since the prior marginal scores are not used to construct a stochastic process directly, and, instead, only to regularize the posterior process at meta-test time. In addition, functional approximate inference techniques such as fSVGD and fBNNs typically use fixed measurement set sizes so that marginal consistency of the prior marginals is not a concern anyhow.
>
> > In 5.2, the authors propose to fit each task by a GP in order to use eq. (2). If you can use a GP to model each task, do you still need the score network? Can we just fit a prior for all GPs? like a Hierarchical GP? Have you tested on such option?
>
> We only use the GPs to independently interpolate the meta-training datasets. This is quite different from fitting a GP prior:
> The stochastic process marginals corresponding to our meta-learned score network can assume arbitrary distributions and be highly non-Gaussian. In contrast, if we meta-learn GP priors, the prior marginals are multivariate Gaussians, which is much more restrictive. Overall, MARS can meta-learn much more flexible and complex stochastic process priors than a GP prior can ever assume.
>
> In response to your inquiry, we have added the baseline ‘PACOH-GP’ which (hierarchically) meta-learns a GP prior to our benchmark study. As the updated results in Tables 1 and 2 indicate, MARS outperforms PACOH-GP in the majority of environments. This suggests that the additional flexibility of MARS, as compared to GPs, helps when meta-learning and representing inductive bias.
>
> In response to your and the other reviewers’ concerns about using GPs, we have also added a MARS-BNN variant to the paper where, instead of a GP, we use an MC-dropout BNNs to interpolate the datasets. Using BNNs as interpolators performs similarly in performance (see updated Tables 1, 2) and allow us to circumvent the scaling issues of a GP.
>
> [1] Hyvärinen, A., & Dayan, P. (2005). Estimation of non-normalized statistical models by score matching. Journal of Machine Learning Research, 6(4).
>
> [2] Sun, S., Zhang, G., Shi, J. and Grosse, R., 2019. Functional variational Bayesian neural networks. arXiv preprint arXiv:1903.05779.

---

### Official Review · Reviewer_VdFm · 2022-10-25

**Confidence:** 4
**Correctness:** 3
**Technical Novelty And Significance:** 3
**Empirical Novelty And Significance:** 3
**Recommendation:** 8

**Clarity, Quality, Novelty And Reproducibility:**

This could be a solid paper if the weaknesses are addressed: The idea is original. The literature review is thorough and mostly accurate. The execution is smooth and the writing is easy to follow.

Some minor issues:
* Section 2 > score estimation > "there is a body of work on nonparameteric score estimation. The KSD paper did not propose a score estimation method; the same applies to the inverse problem book.

* Section 4.3 > Second paragraph > "From this viewpoint, ..., is p(h) a stochastic process on the function space", p(h) should be removed.

**Strength And Weaknesses:**

## Strengths

* The paper leverages ideas from several different areas (meta learning/GPs/BNNs/score estimation) and combines them in a creative way. The review paragraphs are very well-written and clearly convey the author's line of thought.

* The idea of using score models instead of normalized priors for Bayesian meta learning is a clever one and overcomes many challenges faced by existing methods.

* The experimental results are strong, if not extensive, and come with careful ablation study that shows the benefit of each component.

## Weaknesses

* One big limitation I could see from the formulation is that different tasks must share the same input space and output space. I am not an expert in meta learning so not sure if this is a commonly-made assumption, but it does feel significantly limiting the application scope.

* Using a user-specified measurement distribution seems unnatural to me. Why not just rely on the empirical distribution of X from the training data?

* Relying on GPs to interpolate at places where there is no training data is also a bit unsatisfying. I could imagine the GP's behavior will have a great influence on the performance. Could we do an ablation study by remove the GP? Also, it might be difficult to answer this question--why not use a GP prior if we are relying on a GP anyway?

* In the ablation study, it would be nice to compare to the nonparametric score estimator using curl-free kernels (i.e., those based on kernel exponential families) which were shown to have drastically better performance in Zhou et al.

* Some experiment details are missing, e.g., what is the measurement distribution being used in experiments? What kind of algorithms are used to fit the GP? Do you use a sparse approximation or the dataset is small enough?

**Summary Of The Paper:**

This paper proposes a new method for Bayesian meta learning by learning priors in the function space. The key insight that distinguishes this work from prior literature is that functional inference techniques only make use of the prior score function and do not require a normalized prior density; therefore, instead of fitting a restricted form of prior distribution during meta training, one can fit a flexible score model of the function outputs and use it to transfer knowledge to new tasks.

**Summary Of The Review:**

Overall I am positive about this paper and would be happy to raise my score if the rebuttal sufficiently addresses the weakness part. The idea is novel and addresses an important problem in Bayesian meta-learning. The empirical study is very careful and demonstrates the effectiveness of the proposed method.

---

> ### Author Response · Authors · 2022-11-15
> **Authors' response [2/2]**
>
> > In the ablation study, it would be nice to compare to the nonparametric score estimator using curl-free kernels (i.e., those based on kernel exponential families) which were shown to have drastically better performance in Zhou et al.
>
> In Appendix D.1 we compare the performance of several nonparametric score estimators. Our quantitative comparison in Tables 7 and 8 included the kernel exponential family estimators, its Nyström extension (NKEF), Stein estimator, and Spectral Stein Gradient Estimator (SSGE).
>
> Per your suggestion, we have now also added the nonparametric estimators proposed by Zhou et al., i.e., the $\nu$-method as well as the KEF method with conjugate gradients (KEF-CG). In our experiments, we consider both estimators with various curl-free kernels which were used by Zhou et al. (2020). We found that, on average over the four evaluation environments, SSGE performs best among the nonparametric score estimators, both in terms of residual mean squared error and cosine similarity. This is why we use SSGE in the ablation study in Section 6.3.
>
> > Some experiment details are missing, e.g., what is the measurement distribution being used in experiments?
>
> As we specify in Section 5.4., we use a uniform distribution over a hypercube that conservatively covers the data in $\mathcal{X}$. In particular, we construct this hypercube heuristically by taking the minimum and maximum value from the meta-training data in each dimension and expanding the resulting range by 20% to each side. We have added a section to the appendix of the manuscript (i.e. Appendix A.5.) in which we better explain and formalize this.
>
> >What kind of algorithms are used to fit the GP? Do you use a sparse approximation or the dataset is small enough?
>
> In all our experiments the number of meta-training points per dataset never exceeds 300 points. Thus, we just use the closed-form solution of the GP prior. We have added an appendix section with details on how we use the GPs (see Appendix A.4.).
>
>
> We hope that with our answers and newly conducted experiments, we have addressed your concerns. Please let us know if there are any remaining questions or shortcomings in your eyes. If not, would you be willing to increase your score, as suggested in your review?

---

> ### Author Response · Authors · 2022-11-15
> **Authors' response [1/2]**
>
> Thanks a lot for reading the paper so thoroughly and giving detailed feedback. In the following, we respond to your concerns.
>
> > One big limitation I could see from the formulation is that different tasks must share the same input space and output space. I am not an expert in meta learning so not sure if this is a commonly-made assumption, but it does feel significantly limiting the application scope.
>
> That the tasks have the same input space and output space is a very common assumption in meta-learning and transfer more broadly. Indeed, we agree with the reviewer that this limits the application scope, e.g., excludes sequence data of varying lengths. However, there are many meta-learning applications and algorithms where the dimensionality of the data space is the same across tasks. In all these cases, our MARS approach applies without modifications. Our experiment section includes problems in experimental physics, health care, and robotics, thus, showing the breadth of possible applications.
>
> > Using a user-specified measurement distribution seems unnatural to me. Why not just rely on the empirical distribution of X from the training data?
>
> As measurement distribution, we use a uniform distribution over the domain instead of the empirical distribution for two reasons:
>
> First, we not only look at the final accuracy of predictions but also want to obtain good uncertainty estimates. Crucially, the uncertainty estimates should also be calibrated beyond the training data. We can only insure this when we train our score network (see Alg. 1) as well as the functional BNN (see Alg. 2 in Appendix) with measurement sets that go beyond training data.
>
> Second, our score network needs to make accurate predictions for arbitrary measurement sets. If we only train it with the potentially small number of points in the meta-training data, we cannot make sure that it generalizes well to arbitrary measurement sets at meta-test time.
>
> > Relying on GPs to interpolate at places where there is no training data is also a bit unsatisfying. I could imagine the GP's behavior will have a great influence on the performance. Could we do an ablation study by remove the GP?
>
> In general, we require a regression model in order to interpolate each dataset so that we can estimate the marginal scores in arbitrary measurement points. Ideally, this regression model should have epistemic uncertainty estimates so that we can propagate the interpolation uncertainty into the score estimates. We chose GPs for computational convenience (i.e., they have a closed-form solution) and their good uncertainty estimates. However, in place of GPs, we can use many other uncertainty-aware regression models such as ensembles of neural networks (NNs) or any form of Bayesian neural network (BNN).
>
> We agree with your concern about the potential limitations of GPs. Thus, we have conducted experiments where, instead of GPs, we use various NN-based methods to interpolate the datasets. In particular, we have tested NN ensembles and MC-dropout BNNs [1] for this purpose. Since MARS with MC-dropout BNNs performed consistently better than with NN ensembles, we now propose the former as an alternative to GPs in Section 5.2. We refer to this method as MARS-BNN, as opposed to MARS-GP which uses GPs for interpolations.
>
> We have added the corresponding experiment results for MARS-BNN to Tables 1 and 2. In summary, MARS-BNN performs similarly to MARS-GP in terms of the RMSE. In the case of uncertainty calibration, MARS-BNN performs on average a bit worse than MARS-GP, though, still outperforms most other meta-learning approaches. This is not surprising as the MC-dropout uncertainty estimates are not as reliable as those of a GP. However, by using BNNs as uncertainty-aware interpolators, we can avoid scaling issues like in the case of GPs. Overall, we think that this makes the method more versatile and more convincing.
>
> [1] Gal, Y., & Ghahramani, Z. (2016). Dropout as a bayesian approximation: Representing model uncertainty in deep learning. In international conference on machine learning (pp. 1050-1059)
>
> > Also, it might be difficult to answer this question--why not use a GP prior if we are relying on a GP anyway?
>
> The stochastic process marginals corresponding to our meta-learned score network can assume arbitrary distributions and be highly non-Gaussian. In contrast, if we meta-learn Gaussian Processes priors, the prior marginals are multivariate Gaussians, which is much more restrictive. Overall, MARS can meta-learn much more flexible and complex stochastic process priors than a GP prior can ever assume.
>
> To investigate whether this translates into better empirical performance, we have added the baseline ‘PACOH-GP’ which meta-learns a GP prior to our experiments. As the updated results in Tables 1 and 2 indicate, MARS outperforms PACOH-GP in the majority of environments.

---

### Official Review · Reviewer_Wct9 · 2022-11-04

**Confidence:** 4
**Correctness:** 4
**Technical Novelty And Significance:** 3
**Empirical Novelty And Significance:** 3
**Recommendation:** 8

**Clarity, Quality, Novelty And Reproducibility:**

The paper is beautifully written and very clear and well-organized.
I want to highlight this as an achievement since there are many moving parts in this work that are orchestrated in a complex way and the authors writing guides the reader through them lucidly and provides context at each stage.

Further, the approach tackles a popular problem, but uses an interesting combination of techniques and I particularly enjoy the amortization over the score estimate via the attention model as a novel and interesting contribution to approximate inference.

**Strength And Weaknesses:**

Strengths:
- The approach is very elegant conceptually, since the authors strive to overcome the specific problems of meta-learning and functional BNN estimation via score functions jointly and develop an integrated approach which utilizes amortization of the score estimate using a transformer based on a graceful objective.
- the performance of the approach seems to be good in the empirical part of the paper


Weaknesses:
- There is one aspect in this paper that feels dissatisfactory: the fact that the authors require a GP to be learned on their datasets in order to be able to learn a NN, since this inherently assumes the same inductive bias as the GP and the score function estimator will be biased towards the types of interpolations of the GP. It is basically somewhat suspicious that the authors rely on the GP as the key to overcome here but in their model they cannot overcome the GP's limitations in case it is misspecified as it directly is upstream of their score estimator.
- Given that the GP is fit directly on the observations, this also means the authors inherit all the GP limitations in terms of computation and so on, which here of course is only relevant for training and not for testing, but is still somewhat dissatisfactory.


**Summary Of The Paper:**

The authors propose a method for meta learning in function space, denoted MARS.

Their approach cosists of two stages:
first, the authors train a GP on task-specific data to interpolate between different function values.
They then use this to be able to train a score function model given by a transformer which estimates the score of the meta learning model per task.
Second, the authors use their learned score matching network to perform functional BNN inference and meta-learn a prior.

In their experiments the authors show good performance compared to various baselines.


**Summary Of The Review:**

The authors propose an interesting, complicated, and ultimately empriically useful way to perform meta-learning in function space using BNNs.

While the approach has some modeling drawbacks by its dependence during training on GPs which I feel makes the approach somewhat less likely to generalize than I would have hoped, I am very excited about the ideas around amortizing the score estimate and the overall positioning and technical execution here.

As such, I believe this paper has value to the community for its  technical ideas and the strong presentation, even if I have doubts about the core model.

---

> ### Author Response · Authors · 2022-11-15
> **Authors' response**
>
> Thank you very much for the encouraging feedback and helpful comments. In the following, we respond to your main concern: the use of GPs for interpolating the datasets.
>
> The algorithm presented in Section 5 of the paper is one particular instantiation of our general meta-learning approach. In general, we require a regression model in order to interpolate each dataset so that we can estimate the marginal scores in arbitrary measurement points. Ideally, this regression model should have epistemic uncertainty estimates so that we can propagate the interpolation uncertainty into the score estimates.
> We chose GPs for computational convenience (i.e., they have a closed-form solution) and their good uncertainty estimates. However, in place of GPs, we can use many other uncertainty-aware regression models such as ensembles of neural networks (NNs) or any form of Bayesian neural network (BNN).
>
> We agree with your concern about the potential limitations of GPs. Thus, we have conducted experiments where, instead of GPs, we use various NN-based methods to interpolate the datasets. In particular, we have tested NN ensembles and MC-dropout BNNs [1] for this purpose. Since MARS with MC-dropout BNNs performed consistently better than with NN ensembles, we now propose the former as an alternative to GPs in Section 5.2. We refer to this method as MARS-BNN, as opposed to MARS-GP which uses GPs for interpolations.
>
> We have added the corresponding experiment results for MARS-BNN to Tables 1 and 2. In summary, in terms of RMSE, MARS-BNN performs similarly to MARS-GP. In terms of uncertainty calibration, MARS-BNN performs on average a bit worse than MARS-GP, though, still outperforms most other meta-learning approaches. This is not surprising as the MC-dropout uncertainty estimates are not as reliable as those of a GP. However, by using BNNs as uncertainty-aware interpolators, we can avoid scaling issues like in the case of GPs.
>
> Overall, we think that this makes the method more versatile and more convincing. Thanks a lot for the constructive feedback that helped us to improve the paper! What do you think; is the MARS-BNN version more satisfactory since it does not suffer from the scaling issue of GPs? If yes, are you willing to increase your score, now, that your concerns were addressed?
>
> [1] Gal, Y., & Ghahramani, Z. (2016). Dropout as a bayesian approximation: Representing model uncertainty in deep learning. In international conference on machine learning (pp. 1050-1059)

---

> > ### Comment · Reviewer_Wct9 · 2022-11-22
> > **Thank you for the response**
> >
> > Thank you for the update.
> >
> > I am glad you added the variant without a GP, it makes the paper stronger in my opinion, though my core criticism that the inferred function will inherit the "calibration" or "beliefs" of the approximator (whether GP or BNN) remains and would need a deeper response.

---

### Author Response · Authors · 2022-11-15
**Updates to the paper in response to reviews**

In response to the reviewers’ concerns and requests, we have performed additional experiments and made the following updates to the manuscript:
* In addition GPs for interpolating the datasets, we now propose to use MC-dropout BNNs (i.e., MARS-BNN) as an alternative variant in Section 5.2.
* We include the MARS-BNN variant in our benchmark study in Section 6.2. In response to the request of reviewer HXi8, we now also include PACOH-GP, an approach that meta-learns GP priors in Tables 1 and 2.
* We have added details about MARS-BNN variant to Appendix A.4.
* We have added Section A.5 to the appendix where we explain and formalize the choice of measurement distribution in more detail.
* In response to the request of reviewer VdFm, we added nonparametric score estimators with curl-free kernels to the score estimator benchmark in Appendix D.1.
* We have added Section D.4 in the appendix where we empirically investigate the use of further possible interpolation methods such as a single NN or NN ensembles in MARS.

---

### Decision · Program_Chairs · 2023-01-20

**Decision:**

Accept: notable-top-25%

**Justification For Why Not Higher Score:**

The paper seems like a solid contribution, but it lacks the outstanding characteristic required for an oral.

**Justification For Why Not Lower Score:**

The paper got a high average rating of 7.33. All the reviewers are positive.

**Metareview: Summary, Strengths And Weaknesses:**

Summary:

The paper presents a new Bayesian meta learning method. The idea is to use a stochastic process prior for all tasks rather than a diagonal Gaussian prior for Bayesian neural networks. Instead of estimating the prior, the authors propose to only fit the score function by a transformer encoder and then use fSVGD in meta training and test: A GP is fitted for each task in order to generate the values on all possible measurement set and spectral normalization is used to further reduce the tendency of overfitting.

Strengths:

- Elegant approach.
- Good empirical performance.
- Clearly written and very well organized
- idea of using score models instead of normalized priors for Bayesian meta learning is a clever one and overcomes many challenges
- The experimental results show the consistently good performance comparing benchmark methods.

Weaknesses:

- the inferred function will inherit the "calibration" or "beliefs" of the approximator (whether GP or BNN).
- different tasks must share the same input space and output space
- score network architecture itself does not enforce marginal consistency

Recommendation:

All the reviewers vote for acceptance. I therefore recommend to accept the paper and encourage the authors to use the feedback provided to improve the paper for its final version.


**Note From Pc:**

if the above contains the word "oral" or "spotlight" please see: "oral" presentation means -> notable-top-5% and "spotlight" means -> notable-top-25%. As stated in our emails, we are disassociating presentation type from AC recommendations